# Notch activity is modulated by the aGPCR Latrophilin binding the DSL ligand in *C. elegans*

Willem Berend Post [1,10], Victoria Elisabeth Groß[1,10], Daniel Matúš[2,3,10], Iannis Charnay [1], Fabian Liessmann [4], Florian Seufert [5], Peter Hildebrand [5], Jens Meiler[4,6,7], Anette Kaiser [8], Torsten Schöneberg[2,9] & Simone Prömel [1]✉

The Notch pathway is a highly conserved signaling cascade across metazoans that regulates numerous physiological processes, including cell proliferation, differentiation, and fate determination. Given its fundamental roles, the pathway is tightly regulated by diverse molecules through multiple mechanisms. Here, we identify the Adhesion GPCR latrophilin (LPHN/ADGRL) as a positive modulator of Notch signaling, which increases Notch receptor activation and the translocation of its intracellular domain into the nucleus. Physiologically, this latrophilin role is crucial for balancing the number of proliferating cells in the gonadal stem cell niche of the nematode *C. elegans*. In silico, in vitro, and in vivo analyses demonstrate that the *C. elegans* latrophilin homolog LAT-1 directly interacts with the DSL protein/Notch ligand LAG-2 on the same cell. This interaction is mediated by LAT-1's conserved GAIN and the RBL domain. Importantly, the modulatory effect depends solely on the receptor's N terminus and is independent of G protein signaling. Finally, we explore the implications of this fine-tuning of Notch signaling by an aGPCR.

Adhesion *G* protein-coupled receptors (aGPCRs) are unique receptors involved in various (patho-)physiological processes (summarized in refs. 1–3). Beyond transmitting traditional *G* protein-mediated signals[4,5], their large N termini enable them to perform functions independent of canonical signaling (often referred to as *N* terminus-only, *trans*, or seven transmembrane domain (7TM)-independent functions). These include facilitating cell-cell and cell-matrix adhesion[1,6], as well as influencing other signaling pathways. For instance, some aGPCRs have been shown to modulate Wnt signaling[7–10]. This versatility positions aGPCRs as key integrators of diverse physiological signals.

In this study, we present evidence that the aGPCR latrophilin (LAT/LPHN/ADGRL) interacts with the Notch pathway, modulating its activity by enhancing the activation of the Notch receptor. The Notch pathway is one of the most essential signaling cascades in many organisms governing numerous physiological processes such as development and tissue morphogenesis (summarized in refs. 11,12). Due to its fundamental role, many components of the Notch pathway and the underlying mechanisms are highly conserved across species[13–15]. In the nematode *C. elegans*, the Notch pathway controls, among others, germ cell division and the balance between germ cell proliferation (self-renewal) in the gonadal stem cell niche and

[1]Institute of Cell Biology, Department of Biology, Heinrich Heine University Düsseldorf, Düsseldorf, Germany. [2]Rudolf Schönheimer Institute of Biochemistry, Medical Faculty, Leipzig University, Leipzig, Germany. [3]Department of Molecular and Cellular Physiology, Stanford University, Stanford, CA, USA. [4]Institute for Drug Development, Faculty of Medicine, Leipzig University, Leipzig, Germany. [5]Institute of Medical Physics and Biophysics, Medical Faculty, Leipzig University, Leipzig, Germany. [6]Center for Scalable Data Analytics and Artificial Intelligence ScaDS.AI, Leipzig University, Leipzig, Germany. [7]Department of Chemistry, Center for Structural Biology, Vanderbilt University, Nashville, TN, USA. [8]Department of Anaesthesiology and Intensive Care, Medical Faculty, Leipzig University, Leipzig, Germany. [9]School of Medicine, University of Global Health Equity, Kigali, Rwanda. [10]These authors contributed equally: Willem Berend Post, Victoria Elisabeth Groß, Daniel Matúš. ✉e-mail: proemel@uni-duesseldorf.de

differentiation into germ cell progenitors (entry into meiosis). Thereby, it promotes proliferation while suppressing meiotic entry[16–20]. The signal originates from the distal tip cell (DTC), where LAG-2, a membrane-bound Notch ligand of the Delta, Serrate, LAG-2 (DSL) family, interacts with the Notch receptor GLP-1 on adjacent germ cells[19–21] thereby activating it and initiating the Notch pathway.

Here, we show that LAT-1, the *C. elegans* homolog of the aGPCR latrophilin, directly interacts with the Notch ligand LAG-2 and enhances Notch receptor activation. This interaction increases Notch activity solely via the aGPCR N terminus (N terminus-only/*trans*/7TM-independent function) and modulates cell proliferation within the gonadal stem cell niche.

## Results
### LAT-1 modulates germ cell proliferation

Recently, we discovered that the 7TM-independent/N terminus-only/*trans* function of LAT-1 in *C. elegans* regulates various processes, including germ cell apoptosis, ovulation, and sperm guidance,

ultimately controlling reproduction[22]. In our analysis of hermaphrodites homozygous for the null allele *lat-1(ok1465)* (hereafter referred to as *lat-1*), we also observed a reduced number of germ cells in the distal region of the gonad (Fig. 1a, b). This reduction primarily affects the progenitor zone but also extends to the transition zone and the pachytene stage (Fig. 1a, c). Using mNeonGreen-fused REC-8 as a marker for progenitor zone nuclei[23], we confirmed that the size of the progenitor zone of *lat-1* mutants is indeed modestly yet significantly smaller than in wild-type hermaphrodites (Fig. 1d, e). Time-course analyses revealed that these phenotypes were most prominent in worms aged L4 + 8 h (Supplementary Fig. 1a–c). Consequently, all subsequent experiments were conducted on individuals at this stage.

To further characterize the effects of LAT-1 on distal germ cells, cell proliferation was studied by assessing both mitotic (M) phase (using an anti-phospho-histone H3 (Ser10) (PH3) antibody) and synthesis (S) phase (using EdU staining) nuclei. These analyses revealed a decrease in both mitotic (Fig. 1f–h) and DNA synthesis (Fig. 1i–k) events in *lat-1* mutants.

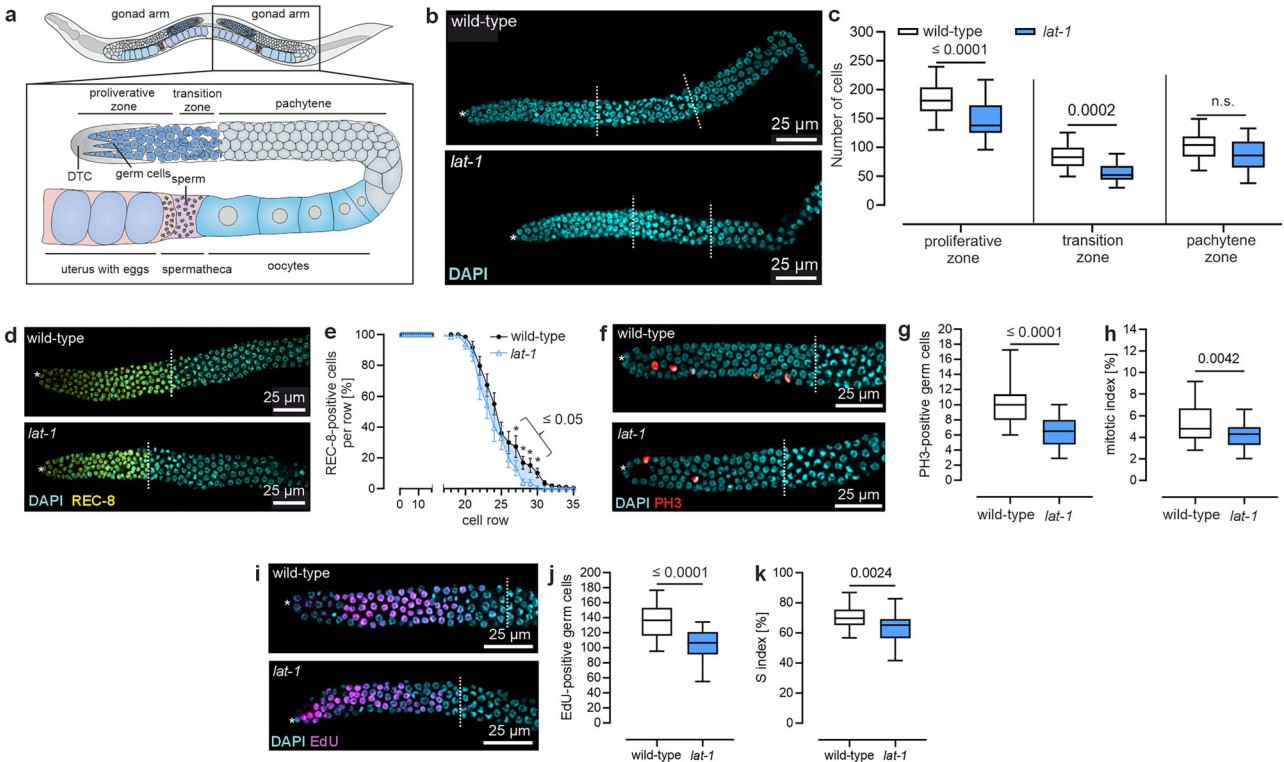

**Fig. 1 | Nematodes lacking *lat-1* display a reduction in germ cells. a** Schematic depiction of the adult *C. elegans* gonad. The nematode has two symmetrical U-shaped gonads where oocytes are produced during adulthood. The most distal part of each gonad arm, the progenitor zone, is encased by the distal tip cell (DTC). Within this zone, germ cells undergo continuous self-renewal through mitotic divisions. Subsequently, they enter meiosis in the transition zone, where they arrest in the pachytene before cellularizing, growing, and progressing further through meiotic development. Eventually, the oocytes are pushed through the spermatheca, where sperm, produced during the last larval stage, is stored. After fertilization, the eggs move to the uterus and are laid. **b** Hermaphrodite gonads lacking LAT-1 display altered germ cell zone sizes. Shown are two representative images of gonads of L4 + 8 h-old hermaphrodites. Asterisks: DTC; dashed lines: zones (**c**) Quantification of germ cell numbers in (**b**) reveals shorter zones in *lat-1* worms. Wild-type: *n* = 33, *lat-1*: *n* = 19 in four independent experiments.
**d** Expression of mNeonGreen-fused *rec-8* ceases in more distal germ cell rows in *lat-1* individuals compared to wild-type controls. Asterisks: DTC; dashed lines: end of progenitor zone. **e** Quantification of REC-8-positive cells per cell row shown in (**d**) confirms the shorter progenitor zone and thus, the earlier change in the transition from progenitor (REC-8-positive cells) to meiotic (REC-8-negative cells) fate

in *lat-1* mutant germlines. Wild-type: *n* = 23, *lat-1*: *n* = 18 in four independent experiments. Exact *p* values are provided in the Source Data. **f** Representative images of PH3-stained gonads of 8 h-post-L4 *lat−1* and wild-type individuals. Asterisks: DTC; dashed lines: end of progenitor zone. **g** PH3-stained germlines from (**f**) reveal a lower incidence of M phase nuclei in the progenitor zone of *lat-1* compared to wild-type gonads. Wild-type: *n* = 44, *lat-1*: *n* = 38 in six independent experiments. **h** *lat-1* mutant germlines have a decreased M index (percentage PH3-positive nuclei from all progenitor zone nuclei). Wild-type: *n* = 44, *lat-1*: *n* = 38 in six independent experiments. **i** EdU-stained gonads 8 h-post-L4 worms visualize S phase nuclei. Asterisks: DTC; dashed lines: end of progenitor zone. **j** Quantification of images from (**i**) demonstrates that the distal germline of *lat-1* hermaphrodites has more EdU-positive (S phase) nuclei than wild-type controls. Wild-type: *n* = 34, *lat-1*: *n* = 26 in 6 independent experiments. **k** Quantification of S indices based on images from (**i**). *lat-1* germlines have a decreased S index (percentage of EdU-positive nuclei from all progenitor zone nuclei). Wild-type: *n* = 34, *lat-1*: *n* = 26 in six independent experiments. Graph raw data are provided in the Source Data. Graph details and statistics are: (**c**), (**g**), (**h**), (**j**), (**k**): Box plots with median (center), interquartal range, 5th (lower whisker) and 95th (upper whisker) percentiles. **e**: Mean ± SEM. Two-sided unpaired t-test without multiple comparison correction.

Additionally, we calculated the M (Fig. 1h) and S (Fig. 1k) indices, defined as the percentage of M phase (PH3-positive) (Fig. 1g, Source data) or S phase (EdU-positive) (Fig. 1j, Source data) nuclei within the total number of nuclei in the progenitor zone. These indices, which can account for potential zone size differences, were lower in *lat-1* mutant compared to wild-type hermaphrodites. The observed effects in *lat-1* mutants were persistent at different temperatures (Supplementary Fig. 1d–f).

Taken together, our results indicate that LAT-1 plays a role in controlling cell proliferation and/or regulating zone size during the late L4/very early adult stage.

### LAT-1 facilitates Notch signaling in the distal *C. elegans* germline

Various potentially coexisting mechanisms could lead to a reduction in the size of the germline zones, such as a smaller pool of mitotic cells or an altered cell cycle (reviewed in[24]). It is possible that LAT-1 plays a role in one or more of these processes. A key signaling cascade that promotes germ cell proliferation while suppressing meiotic entry, thereby regulating the balance between the two and determining the size of the progenitor zone, is the Notch pathway. Some, albeit by far not all, of the defects in *lat-1* nematodes resemble phenotypes reported in mutants of these Notch pathway components. For instance, a decreased number of proliferating cells and an altered progenitor zone size have been observed in *glp-1* reduction-of-function mutants[16,25], leading us to investigate whether LAT-1 affects the Notch pathway. In the *C. elegans* stem cell niche, the initial signal in this highly conserved cascade originates from the DTC, which expresses the DSL ligand *lag-2*. The interaction of this membrane-bound molecule with the Notch receptor GLP-1 on adjacent germ cells triggers the activation of GLP-1, which is cleaved twice with one cleavage by a γ secretase yielding the release of the Notch intracellular domain (NICD). This NICD acts as a transcription factor for the direct transcriptional targets (*lst-1/sygl-1*), which in turn repress *gld-1-3* and other mitosis-inhibiting or meiosis-promoting factors (Fig. 2a) (summarized in[26,27]).

To examine a potential interplay of *lat-1* with Notch, we first analyzed its downstream target *gld-1*, which inhibits cell proliferation[26,28], in *lat-1* mutants using a GLD-1::GFP reporter. In wild-type hermaphrodites, *gld-1* expression begins and intensifies as the repressing effect of the Notch signal diminishes at a certain distance from the DTC[29]. In contrast, *lat-1* mutants showed a significant upregulation of *gld-1* in more distal germ cells, starting from germ cell row ~14 (Fig. 2b, c). This suggests that LAT-1 influences downstream effectors of the Notch pathway. While GLD-1 is a known downstream target of Notch signaling, research indicates that it can also be regulated by other molecules, particularly those involved in cell cycle control[30].

To investigate whether LAT-1 operates within the Notch pathway, we analyzed genetic interactions between *lat-1* and the core Notch pathway components *lag-2* and *glp-1*. Similar to *lat-1* mutants, hermaphrodites carrying single reduction-of-function mutations in *lag-2* (*lag-2(q420)*[21]) and *glp-1* (*glp-1(bn18)*[31]) exhibit defects in germ cell proliferation and meiotic entry, resulting in a shorter progenitor zone (Fig. 2d) and concomitantly, reduced numbers of M phase nuclei (Fig. 2e, Supplementary Fig. 2a)[25,32,33]. The respective *lat-1; lag-2* and *lat-1; glp-1* double mutants did not show any difference in effect compared to the respective Notch component single mutants (Fig. 2d, e). No additive effects were observed suggesting that *lat-1* might act through key Notch signaling components rather than via another, parallel pathway. To further study the role of LAT-1, we examined a potential direct impact of the aGPCR on the Notch pathway. Upon ligand activation, the Notch receptor undergoes proteolytic processing, allowing for the translocation of its intracellular domain (NICD) to the nucleus, where it promotes the transcription of target genes (see above). Notch activation can be quantified by comparing the ratio of membrane-bound GLP-1 to GLP-1 within germ cell nuclei[34,35]. In *lat-1* nematodes, a

decrease in nuclear GLP-1 was observed (Fig. 2f, g), suggesting that the aGPCR impacts Notch signaling. Meanwhile, overall expression levels of both *lag-2* and *glp-1* were unaffected by the absence of *lat-1* (Supplementary Fig. 2b–e).

As many functions of LAT-1 in the germline have been shown to depend solely on its *N* terminus (anchored to the membrane) rather than on G protein signaling[22], we tested whether this was also true for its role in Notch activation. Indeed, expressing only the *lat-1* N terminus tethered to the membrane via its first transmembrane domain (LNT) resulted in similar levels of Notch activation to those observed in wild-type worms (Fig. 2f, g). This confirms that the N terminus alone is sufficient to mediate LAT-1's effect on Notch activation.

Notch signaling is not only required in the germline, it is essential for numerous other processes in the adult nematode. Thus, we asked whether LAT-1 might also be involved in these contexts. As such, during development, Notch signaling is (among others) essential for the formation of the rectum/anus. *lag-2* mutant nematodes do not form a proper rectum/anus, a function that seems also dependent on *glp-1* and the other Notch receptor *lin-12*[20]. As *lat-1* has been shown to be expressed widely during development[36], we tested whether it might interact with Notch signaling here. We observed that *lat-1* mutant animals show anus defects comparable to *lag-2* mutants, albeit at a lower frequency, while the *lat-1; lag-2* double mutant was indistinguishable from the *lag-2* single mutant (Fig. 2h, i). Furthermore, we elucidated a potential interaction of *lat-1* with the Notch pathway in the nervous system where Notch signals control chemosensory avoidance of octanol[37]. As *lat-1* is also expressed on neurons[38], an interaction with *lag-2* in this context is also conceivable. Therefore, we assessed the avoidance behavior of *lat-1* mutants and found that these animals have a similarly delayed response in octanol avoidance compared to *lag-2* mutants, with double mutant animals showing no increased avoidance compared to single mutants (Fig. 2j). Thus, in both contexts, *lat-1* seems to interact with the Notch pathway hinting towards a wider involvement of LAT-1 in Notch signaling within *C. elegans* physiology.

In summary, these results show that LAT-1 cross-talks with the Notch pathway by modulating Notch receptor activity and regulating *gld-1* expression.

### LAT-1 directly interacts with the Notch ligand LAG-2 via its RBL and GAIN domains

We next asked how LAT-1 exerts its modulatory effects on Notch-pathway activation. The primary source of the Notch signal controlling cell proliferation in the progenitor region is the DTC (Figs. 1a, 2a)[26], and its integrity is crucial for proper signaling. To determine whether the DTC morphology is affected in *lat-1* mutants, we examined the structure of this cell in hermaphrodites 8 h post-L4 stage. Both the plexus and cap length appeared similar to those in wild-type nematodes (Supplementary Fig. 3) indicating that the effect of LAT-1 on the Notch pathway is not caused by alterations in DTC morphology.

As the aGPCR is present on both the DTC and potentially germ cells[22], it is conceivable that it modulates Notch activity by directly interacting with the Notch receptor or ligand. To test this hypothesis, we predicted and analyzed in silico the potential interactions between the extracellular domains of LAT-1 with GLP-1 and LAG-2, respectively, using AlphaFold2 Multimer[39]. Based on the consistent domain-level contacts observed across multiple predicted structures and various domain combinations, the computational model supported a binding hypothesis with interactions between LAT-1 and LAG-2, but no convergence of binding interfaces between LAT-1 and GLP-1 was observed. The interaction with LAG-2 is mediated via the LAT-1 rhamnose-binding lectin (RBL) and GPCR autoproteolysis-inducing (GAIN) domains (Fig. 3a, b, Supplementary Fig. 4). We analyzed the AlphaFold2 Multimer-predicted complex using Rosetta and its interface analysis tools to further characterize the predicted LAT-1/LAG-2 binding interface. The total buried surface area at the interface was

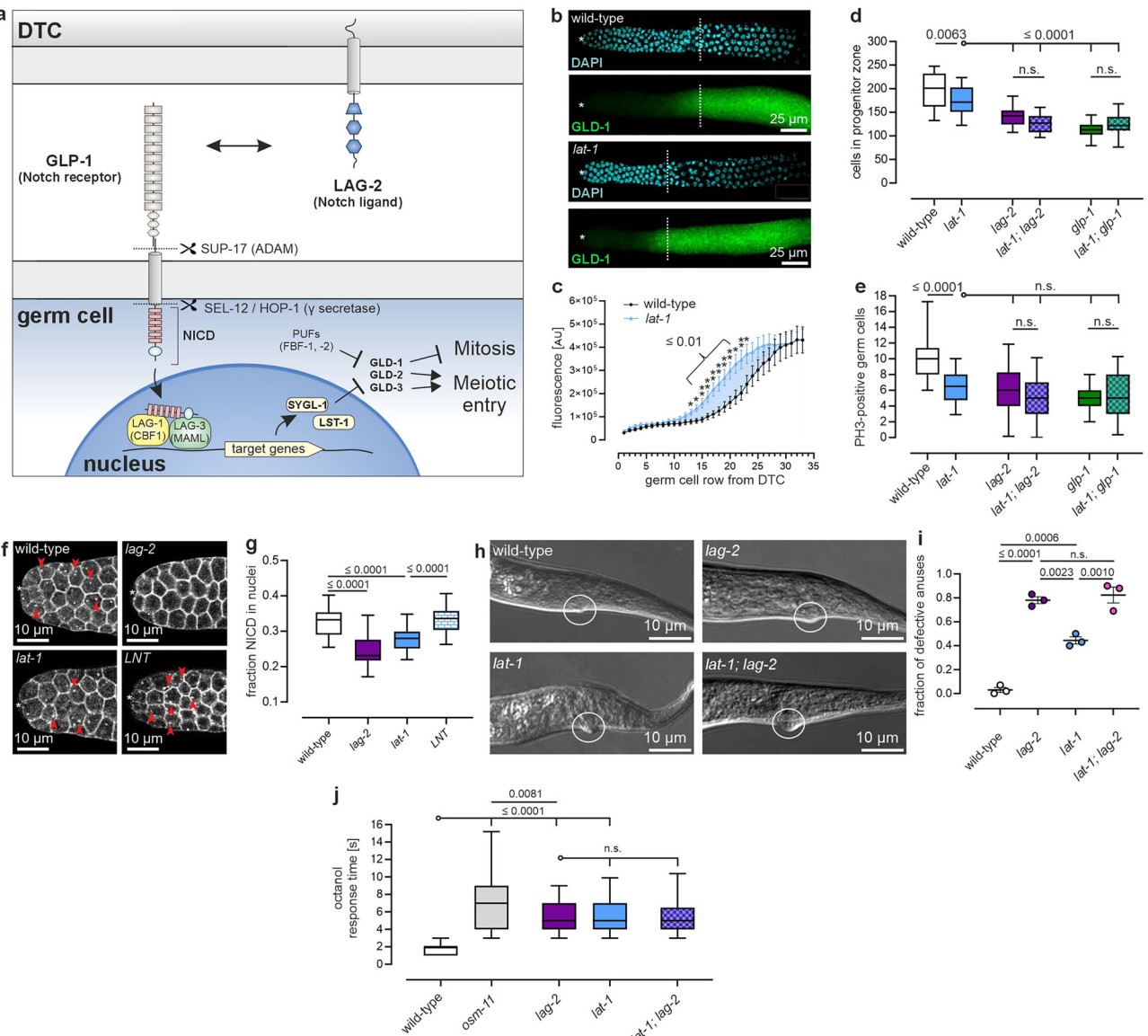

**Fig. 2 | LAT-1 modulates Notch signaling in the distal gonad. a** In the distal gonad, the Notch pathway controls cell proliferation and the switch between germ cell division and differentiation. It is initiated by the Notch ligand LAG-2 on the distal tip cell (DTC) binding to the Notch receptor GLP-1 present on germ cell membranes. This causes the Notch intracellular domain (NICD) to be cleaved off to translocate to the nucleus, initiating together with LAG-1 and LAG-3 the transcription of downstream effectors such as LST-1 and SYGL-1. In turn, these effectors repress expression of e.g., *gld-1-3*. **b** Expression of *gld-1* occurs more distally in *lat-1* mutant (bottom) than in wild-type (top) L4 + 8 h gonads. Asterisks: DTC, dashed lines: end of the progenitor zone. **c** Quantification of *gld-1::GFP* expression/germ cell row in *lat-1* and wild-type gonads by measuring fluorescence intensity with increasing distance from the distal tip cell based on images from (**b**). Loss of *lat-1* leads to a reduced repression of *gld-1*. Wild-type: *n* = 17, *lat-1*: *n* = 22 in four independent experiments. **d, e** Comparison of progenitor zones of Notch pathway component single mutants (*lag-2(q420)*, *glp-1(bn18)*) and respective double mutants with *lat-1(ok1465)*. Analyses are based on DAPI-/PH3-stained gonads to visualize all and specifically M phase nuclei (for images see Supplementary Fig. 2a). The progenitor zone sizes in the double mutants differ from those in *lat-1* single mutants, but are similar to the respective Notch component single mutants (**d**). Overall, no differences in PH3-positive germ cell number were observed in single compared to double mutants (**e**). Replicate values: wild-type: 44,

*lat-1*: 38, *lag-2*: 42, *lat-1; lag-2*: 36, *glp-1*: 41, *lat-1; glp-1*: 46 gonads in 6 independent experiments. **f** Notch activation was visualized using the GLP-1 NICD::V5 reporter *glp-1(q1000[glp-1::4xV5])*. In *lat-1* germline nuclei of L4 + 8 h-old hermaphrodites, less NICD is present than in the wild-type (red arrowheads) which is ameliorated by the presence of the *LNT*. The *lag-2(q420)* mutant (positive control) shows severely reduced NICD in cell nuclei. asterisks: DTC. **g** Quantification of the NICD fraction located in germ cell nuclei based on images in (**f**) confirms the reduced activity of Notch in *lat-1* mutants. Replicate values (independent experiments): wild-type: 33 (5), *lat-1*: 33 (3), *lag-2*: 21 (4), LNT: 16 (3). **h** DIC images showing anus morphology defects in L1 nematodes, which occurs both in *lat-1* and *lag-2* mutants. **i** Quantification of the images from (**h**). Wild-type: 127, *lag-2*: 122, *lat-1*: 167, *lat-1; lag-2*: 86 in three independent experiments. **j** *lat-1* mutant nematodes exhibit a similar delayed reversal upon exposure to octanol as *lag-2* mutants. Replicate values (independent experiments): wild-type: 75 (5), *osm-11*: 45 (3), *lat-1*: 75 (6), *lag-2*: 45 (3), *lat-1, lag-2*: 45 (3). Graph raw data are provided in the Source Data. Graph details and statistics are: (**c**): Mean *gld-1::GFP* expression/germ cell row ± SEM (replicates and exact p values in the Source Data). Two-sided unpaired t-test without multiple comparison correction. (**d**), (**e**), (**g**), (**i**), (**j**): Box plots with median (center), interquartal range, 5th (lower whisker) and 95th (upper whisker) percentiles. One-way ANOVA with Bonferroni post-hoc test. (**i**): Mean ± SEM. One-way ANOVA with Bonferroni post-hoc test.

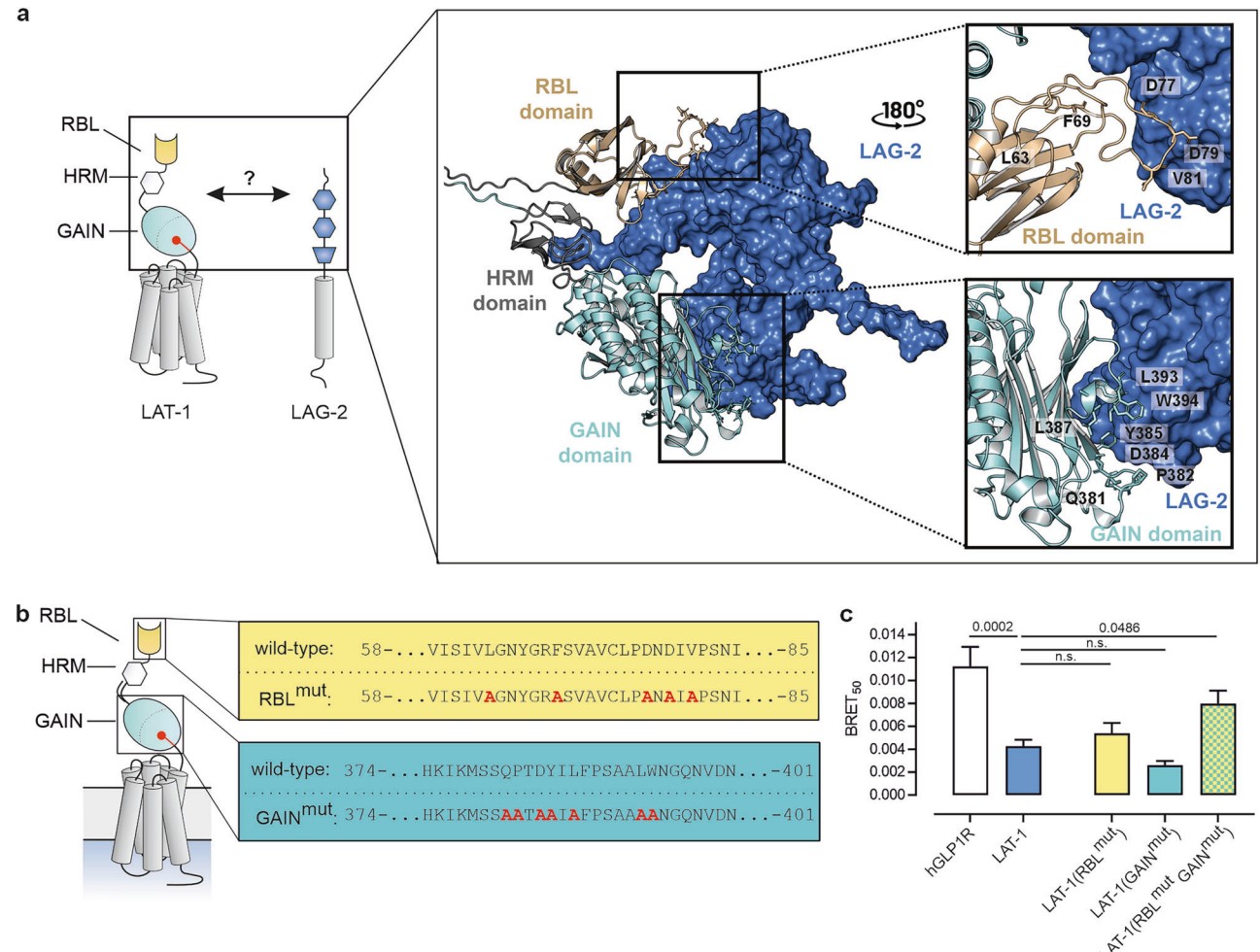

**Fig. 3 | LAT-1 interacts with the Notch ligand LAG-2 in silico and in vitro.**
**a** AlphaFold2 Multimer models suggest an interaction between the extracellular region of LAG-2 (blue, surface representation) and both the RBL (yellow, cartoon representation) and GAIN (cyan, cartoon representation) domains of LAT-1. The interacting regions in LAT-1 include the main protruding loop of the RBL domain and a stretch of the GAIN domain. The HRM domain (gray, cartoon representation) is not involved in binding. Interacting residues are shown as stick representation. Details on energy breakdown values calculated using Rosetta's *per_residue_energies* application are given in Supplementary Fig. 4. **b** Location of the residues within RBL and GAIN domain of LAT-1 identified by models and energetic breakdown analyses (Supplementary Fig. 4) to be candidates to mediate the binding to LAG-2. **c** Nano

Bioluminescence Resonance Energy Transfer (NanoBRET) analyses show an interaction of LAT-1 (Venus::LAT-1) with LAG-2 (Nluc::LAG-2) in HEK293 cells. The human GLP1R fused to Venus served as negative control. Note that higher $BRET_{50}$ values correspond to low binding affinity. Introduction of distinct point mutations in single domains predicted to be essential for the interaction (identified in (**a**–**c**)) do not significantly reduce affinity of LAT-1 for LAG-2, however, affinity is decreased when mutations in RBL and GAIN domain are combined. Mean ± SEM, $n = 4$ technical replicates in 11 (LAT-1), 8 (hGLP1R), 4 (LAT-1 (RBL$^{mut}$)), 4 (LAT-1 (GAIN$^{mut}$)), 10 (LAT-1 (RBL$^{mut}$ GAIN$^{mut}$)) independent experiments. One-way ANOVA with Bonferroni post-hoc test. Corresponding netBRET values and curves are shown in Supplementary Fig. 7b.

approximately 2768 Å², involving contributions from both the RBL and GAIN domain of LAT-1. Notably, the GAIN domain accounts for a larger portion of this interface, contributing 2089 Å². According to Rosetta InterfaceAnalyzer calculations, the interface consists of ~42% polar and 58% hydrophobic contacts. Several predicted hydrogen bonds contribute to complex stabilization, with six of these located within the GAIN domain. These findings support a mixed-mode interaction in which both specificity and stability are likely mediated by a combination of electrostatic and hydrophobic contacts.

To validate the AlphaFold2 Multimer-predicted complex and to investigate its conformational stability, we carried out all-atom molecular dynamics (MD) simulations. The structural integrity of the full LAG-2/GAIN-RBL complex was retained in all three repeats of 1-microsecond MD simulations. Within the simulation time, besides some thermal fluctuations of the GAIN and RBL domains against LAG-2, we did not observe major structural re-arrangements, validating the predicted complex (Supplementary Fig. 5).

Further analysis of these models and the per-residue energy breakdown of the interaction identified five amino acids in the RBL and seven in the GAIN domain to be crucial for the interaction (Fig. 3b, Supplementary Fig. 4). An evolutionary analysis showed that residues forming the binding interface are generally highly conserved among *Caenorhabditis* Spp. (Supplementary Fig. 6) potentially indicating that the interation is also conserved.

We next tested whether the predicted interaction between LAG-2 and LAT-1 indeed occurs in a cellular context. For this purpose, we conducted Nano Bioluminescence Resonance Energy Transfer (Nano-BRET) experiments in vitro. In this setup, LAG-2 fused to a Nanoluciferase (Nluc::LAG-2) served as energy donor while Venus::LAT-1 was the acceptor. Both tags were in the extracellular *N* termini of the proteins. Constructs were expressed heterologously in HEK293 cells (Supplementary Fig. 7a). As negative control that does not interact with LAG-2, the human glucagon-like peptide-1 receptor (hGLP1R) fused to a Venus protein was employed. The NanoBRET analyses confirmed the

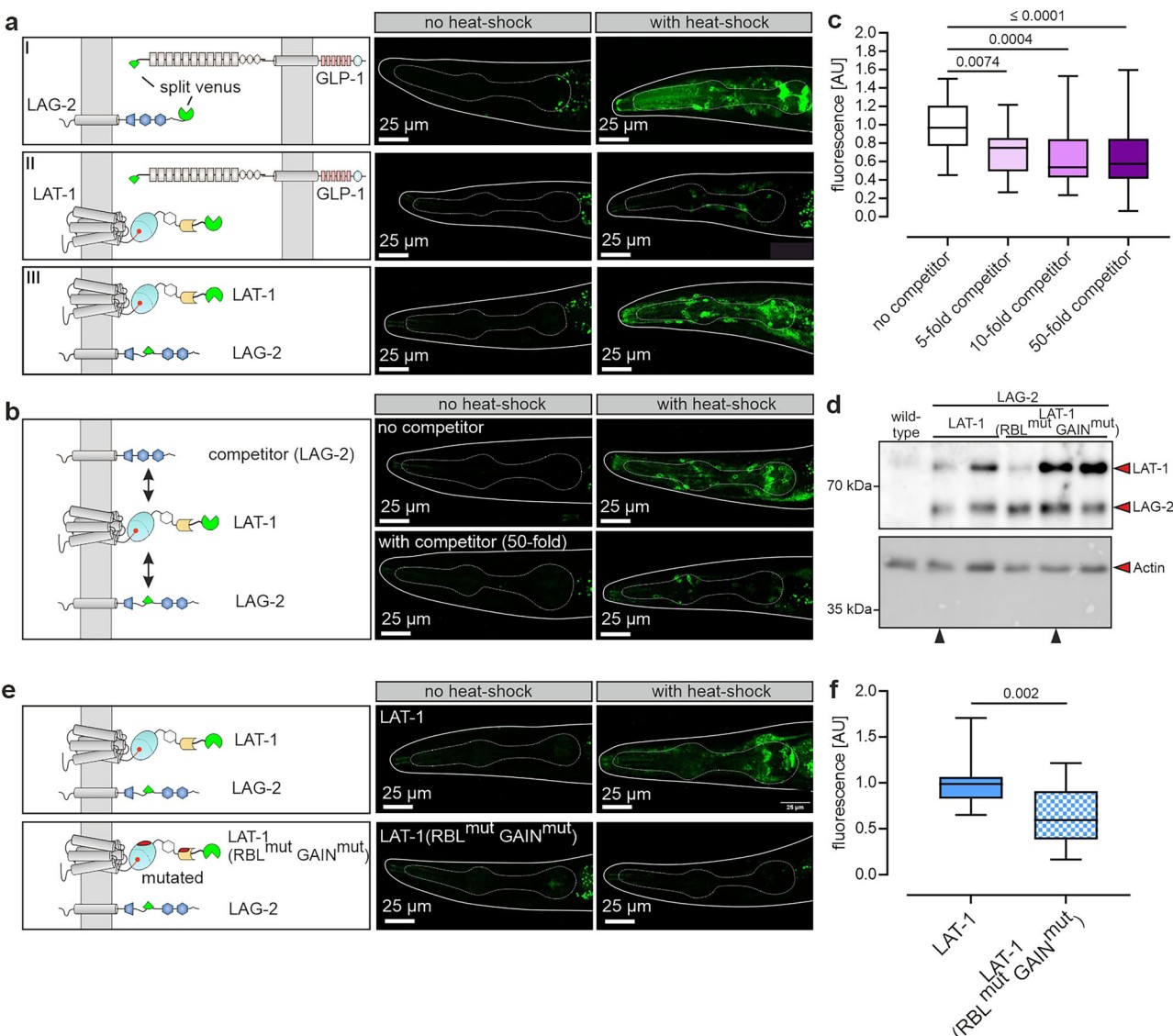

**Fig. 4 | LAT-1 binds to LAG-2 in vivo. a**, **b**, **e** Bimolecular fluorescence complementation (BiFC) analyses using the two parts of a split Venus protein (VN/VC) fused each to a protein of interest in *C. elegans*. Expression of these constructs is driven by the heat-shock promoter *hsp16.41p*. Upon heat shock, the proteins are produced, and a fluorescent signal occurs if they interact as the split Venus protein parts get into proximity, reconstitute, and fluoresce. For signal monitoring, the pharynx was chosen as the heat-shock promoter is highly active in pharyngeal neurons[43]. **a** LAG-2 binds to GLP-1 yielding a clear fluorescence signal after heat shock (I). Co-expression of *VN::lat-1* and *VC::glp-1* shows a very minor fluorescence signal, indicating that they do not interact (II). A strain harboring *VN::lat-1* and *VC::lag-2* together shows fluorescence after heat shock, indicating binding of the two (III). Shown are representative images of 22 (LAG-2; GLP-1), 35 (GLP-1; LAT-1) and 75 (LAT-1; LAG-2) heat-shocked worms. **b** Competition BiFC assay. In the strain expressing *VN::lat-1* and *VC::lag-2* (employed in (**a**) III), varying amounts of a construct containing an untagged *lag-2* serving as a competitor were introduced. Nematodes stably carrying all three constructs were heat-shocked, and fluorescence was measured. Hermaphrodites with 50-fold excess of untagged *lag-2* competitor relative to *VC::lag-2* (1 ng *VC::lag-2*/ 50 ng untagged *lag-2*) displayed severely less fluorescence. This indicates that LAT-1 interacts with the untagged LAG-2, which competes with VC::LAG-2. Shown are exemplary images. **c** Quantification of the competition BiFC assay from images shown in (**b**). Fluorescence significantly decreases with increasing amounts of untagged *lag-2* as a competitor. Four nematode lines carrying fixed amounts of *VN::lat-1* (1 ng) and

*VC::lag-2* (1 ng) together with no competitor (same line as in (**a**) III), 5-fold (5 ng), 10-fold (10 ng), and 50-fold (50 ng) competitor, respectively, were used. To always maintain the same total amount of DNA, pBluescript was supplemented. Replicate values (independent experiments): no competitor: 75 (8), 5-fold: 17 (3), 10-fold: 21 (3), 50-fold: 24 (3). **d** Western Blot analysis confirms the expression of *lat-1* (V5-tagged, 81 kDa (autocatalytically cleaved)), *lag-2* (HA-tagged, 57 kDa), and mutated *lat-1(RBL^mut GAIN^mut)* (V5-tagged, 81 kDa (autocatalytically cleaved)) on protein level in worm lines after heat shock and shows that the point mutations in LAT-1 do not hamper expression. Black arrowheads indicate the lines studied further in (**e**), (**f**). Actin served as a loading control. For full Western blots, see Supplementary Fig. 7e. Western Blot was performed twice with 60-80 worms per sample. **e** Exemplary images of BiFC using VC::LAG-2 and LAT-1(RBL^mut GAIN^mut)::VN showing less fluorescence than in combination with a wild-type VN::LAT-1. **f** BiFC analysis using VC::LAG-2 in combination with VN::LAT-1 carrying the 12 point mutations of residues within the RBL/GAIN domains potentially essential for LAG-2 binding (Fig. 3b). Worm lines selected based on similar LAG-2, LAT-1, and LAT−1(RBL^mut GAIN^mut) protein levels (d, black arrowheads) were tested. The mutations lead to significantly reduced fluorescence levels compared to wild-type LAT-1. Replicate values (independent experiments): LAT-1: 26 (4), LAT-1 (RBL^mut GAIN^mut): 17 (3). Graph raw data are provided in the Source Data. Graph details and statistics are: (**c**), (**f**): Box plots with median (center), interquartal range, 5th (lower whisker) and 95th (upper whisker) percentiles. One-way ANOVA with Bonferroni post-hoc test (**c**). Two-sided unpaired t-test without multiple comparison correction (**f**).

proposed LAT-1/LAG-2 interaction, with BRET$_{50}$ values showing a significantly higher affinity of LAG-2 to LAT-1 than to the hGLP1R negative control (Fig. 3c, Supplementary Fig. 7b).

To further investigate the observed LAT-1/LAG-2 interaction, the amino acids in the RBL and GAIN domain of LAT-1 predicted to mediate the contact interface to LAG-2 were mutated to alanine (Fig. 3b). We verified that all mutated receptors remained intact and cell surface expression was similar to the wild-type (Supplementary Fig. 7c). Mutations in the GAIN domain alone did not affect protein-protein affinity (BRET$_{50}$) in the NanoBRET assay, while mutations in the RBL caused a slight, yet insignificant reduction (Fig. 3c). However, combining mutations in both the RBL and GAIN domains (LAT-1(RBL$^{mut}$ GAIN$^{mut}$)) resulted in a significant two-fold decrease in affinity in the NanoBRET assays compared to the wild-type receptor (Fig. 3c), suggesting that LAT-1 indeed binds to LAG-2 via the RBL and the GAIN domains. It should be noted that a LAT-1 interaction with GLP-1 was not further investigated in this setting due to a lack in *glp-1* expression (see above).

In *C. elegans*, LAG-2 is only one of several Notch ligands. One of the other ligands, APX-1, is so closely related to LAG-2 that it can even compensate for its function in some contexts[40]. Thus, we tested whether APX-1 could have a similar affinity to LAT-1 as LAG-2. Upon expression of *Nluc::apx-1* with *Venus::lat-1*, low BRET$_{50}$ values showed that LAT-1 also has an affinity for APX-1 (Supplementary Fig. 7d), suggesting that LAT-1 may engage in a broader interaction mechanism with several Notch ligands. Notably, the five amino acids within the LAT-1 RBL domain and the seven within the GAIN domain, which are critical for LAG-2 binding (Fig. 3), also appear to contribute to the interaction with APX-1 (Supplementary Fig. 7d). This indicates that the mechanism underlying the binding might be a general one.

### LAT-1 binds the Notch ligand LAG-2 in vivo

To determine whether the observed LAT-1/LAG-2 interaction occurs in vivo in *C. elegans*, Bimolecular Fluorescence Complementation (BiFC)[41,42] was conducted. For that purpose, the two parts of a split Venus protein (VN/VC) were fused to one of the interaction partners, each under the control of the heat-shock promoter *hsp16.41p*. Upon heat shock, the proteins are produced, and a fluorescent signal occurs if they are in close enough proximity to interact. As the heat-shock promotor is particularly strong in pharyngeal neurons[43], both proteins express at levels sufficient to allow detection of even weak interactions. Thus, we assessed the pharynx region to monitor potential BiFC signals. As a positive control, GLP-1 and LAG-2 were used, showing a clear signal upon heat shock (Fig. 4a I). Only very low signals were detected when assessing GLP-1 and LAT-1 (Fig. 4a II), adding to the in silico prediction that they might not interact. Distinct fluorescence was observed in strains carrying LAT-1 and LAG-2 (Fig. 4a III), suggesting that their interaction also occurs in an in vivo context.

It is important to note that, while all other in vivo experiments involving modified *lat-1* in this study rely on genomic editing, this particular assay used extrachromosomal arrays and heat-shock–induced overexpression for two main reasons: (1) endogenous LAT-1/LAG-2 interaction levels are likely too low to be detected, and (2) the reconstitution of the split Venus protein is nearly irreversible[44], potentially leading to stabilized interactions with detrimental effects, especially during development where both LAT-1[36] and Notch molecules[45–47] have important functions.

To ensure that the observed BiFC interaction was specific and not due to a spontaneous reconstitution of the split Venus fluorophores upon overexpression, competition BiFC was conducted[48,49]. In this assay, *VN::lat-1* and *VC::lag-2* were co-expressed with increasing concentrations of an untagged *lag-2* as competitor (5-fold, 10-fold, and 50-fold compared to *VC::lag-2*). If the interaction was specific, the fluorescent signal would decrease since the unlabeled LAG-2 competed with the labeled version. Indeed, a reduction in fluorescence was

detected starting at 5-fold amount of competitor (Fig. 4b, c), indicating the specificity of the interaction.

Next, we tested whether the interaction is mediated via the RBL and GAIN domains in vivo as revealed in silico and in cell culture (see above). Therefore, the LAT-1 construct containing all residues predicted to be essential for LAG-2 binding mutated to alanine residues (LAT-1(RBL$^{mut}$ GAIN$^{mut}$)) was analyzed. First, the expression of this construct was verified and found to be consistent, with similar levels observed across several independently generated worm lines (Fig. 4d). BiFC analyses were performed for one line containing LAT-1(RBL$^{mut}$ GAIN$^{mut}$) and LAG-2 at comparable protein levels (Fig. 4e, Supplementary Fig. 7e). Consistent with the in vitro results, LAT-1(RBL$^{mut}$ GAIN$^{mut}$) showed a reduction in affinity to LAG-2 in vivo as indicated by a decrease of BiFC fluorescence (Fig. 4e, f).

These data show that LAT-1 positively modulates Notch signaling by directly interacting with the Notch ligand LAG-2 in vitro and in vivo. This LAT-1/LAG-2 binding appears to occur via the RBL and the GAIN domains.

### LAT-1 functions from the DTC on neighboring germ cells

Our results indicate that LAT-1 increases Notch activation by binding to the ligand LAG-2 and that only the extracellular N terminus is essential for this, suggesting a *trans* function of the receptor. As LAT-1 is present on the DTC but also on germ cells[22], this raises the question of whether the LAT-1/LAG-2 interaction occurs *in cis* on the DTC or in trans with LAT-1 on the germ cells. To address this, we separately expressed *Nluc::lag-2* and *Venus::lat-1* in different populations of HEK293 cells and varied their ratio as well as the strength of *Venus::lat-1* expression to obtain a protein gradient similar to the experiments described above. The subsequent NanoBRET analyses revealed that with an increasing number of cells expressing *lat-1::Venus*, no NanoBRET window could be established (Fig. 5a), suggesting that a *cis* rather than a *trans* interaction occured. Similarly, increasing the protein expression in the *lat-1::Venus* expressing cells failed to induce a BRET window (Fig. 5a), further strengthening the hypothesis that the interaction forms in *cis*.

As these in vitro data suggest that the LAT-1/LAG-2 interaction occurs on the same cell, this would mean that LAT-1 must be present on the DTC rather than on germ cells in *C. elegans*. To investigate whether this is the case in vivo, two different worm strains were created: one with *lat-1* expressed only in the DTC by using the DTC-specific *lag-2* promoter (*lag-2p::lat-1*) and another with the receptor expressed exclusively in germ cells, driven by the germ cell promoter of *mex-5* (*mex-5p::lat-1*). Both strains were assessed for the size of the progenitor zone and germ cell proliferation through PH3/DAPI staining. When LAT-1 was present in the DTC, progenitor zone size, number of PH3-positive cells, and mitotic index were similar to those observed in wild-type hermaphrodites (Fig. 5b–e). In contrast, when the receptor was expressed in germ cells, zone sizes, PH3-positive cell counts, and mitotic index resembled those found in *lat-1* mutant gonads (Fig. 5b–e). These data, taken together with the in vitro data, indicate that LAT-1 located on the DTC fulfills the receptor's role in germ cell proliferation.

In summary, our findings suggest that LAT-1 interacts with LAG-2 on the same cell, and that LAT-1 must be localized to the DTC, not the germ cells, to modulate Notch signaling and regulate germ cell proliferation (Fig. 5f).

## Discussion

The Notch pathway is a highly conserved signaling mechanism[50] with vital roles in numerous biological processes, especially in development (summarized in refs. 11,51). Unsurprisingly, dysregulated Notch signaling is associated with various diseases, including cancer[52–55]. As such, a tightly regulated network of components exists to control Notch signals (summarized in refs. 56–58). In the present study, we identify the aGPCR latrophilin-1 as a positive modulator of the Notch

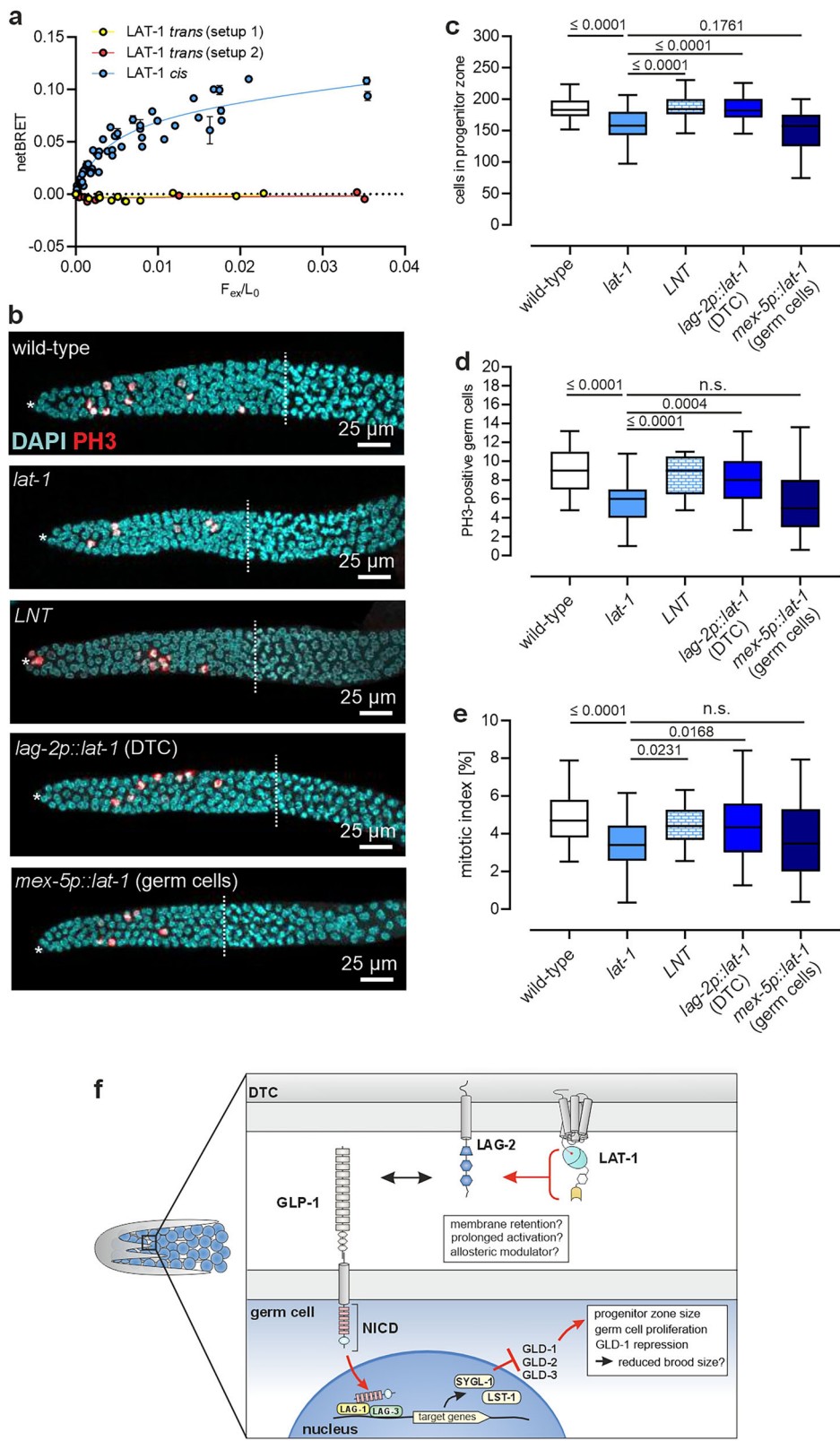

pathway (Fig. 5f). These highly versatile receptors not only transduce G protein-mediated signals into cells like classical GPCR, but also mediate functions independent of *G* protein signals solely via their extracellular *N* termini[59–62]. Our data show that the latrophilin homolog LAT-1 in *C. elegans* enhances the activation/activity of Notch signaling in the germline (Fig. 2). Specifically, the aGPCR increases the translocation of the Notch receptor GLP-1 intracellular domain (NICD) into the nuclei of

distal germ cells, without affecting the overall GLP-1 levels. Once in the nucleus, the NICD functions as a transcription factor, promoting the expression of target genes, thereby regulating the balance between cell proliferation and differentiation into germ cell progenitors (i.e., entry into meiosis) in the distal germline (summarized in ref. 26). In *C. elegans*, two direct target genes of Notch are *sygl-1* and *lst-1*[63], which act in conjunction with the RNA-binding proteins FBF-1 and FBF-2 to

**Fig. 5 | LAT-1 elicits its effect on germ cells from the DTC. a** BRET analysis indicates that LAT-1 and LAG-2 establish their interaction on the same cell (*cis*) (blue line), whereas expression on opposite cells (*trans*) results in no measurable BRET window (red and yellow lines). *n* = 4 technical replicates in 2 (setup 1), 2 (setup 2) and 11 (LAT-1 *cis*) independent experiments. Note that the *cis* BRET curve is the same as in Supplementary Fig. 7b. **b** Representative images of gonads from L4 + 8 h-old hermaphrodites with tissue-specific expression of *lat-1* in the DTC and germ cells, respectively. Only *lat-1* expression in the DTC but not the germ cells resulted in a higher PH3-positive germ cell number and progenitor zone size than in *lat-1* mutants, comparable to the wild-type. As a positive control, *lat-1* promoter-driven LNT was used that ameliorates the reproductive defects in *lat-1* mutants to wild-type levels. Asterisks: DTC, dashed lines: end of the progenitor zone. **c** Quantification of the progenitor zone (from images shown in (**b**)). Replicate values (independent experiments): wild-type: 71 (9), *lat-1*: 53 (12), *LNT*: 37 (7), *lag-*

*2p::lat-1*: 46 (7), *mex-Sp::lat-1*: 30 (8). **d** Quantification of PH3-positive germ cells of images shown in (**b**). Wild-type 71 (9), *lat-1*: 53 (12), *LNT*: 37 (7), *lag-2p::lat-1:* 46 (7), *mex-Sp::lat–1*: 31 (8). **e** M index (percentage PH3-positive nuclei from all progenitor zone nuclei). Wild-type 71 (9), *lat-1*: 53 (12), *LNT*: 37 (7), *lag-2p::lat-1*: 46 (7), *mex-Sp::lat-1*: 30 (8). Raw data of PH3-positive cell counts and the denominators for index calculations are given in the Source data. **f** Proposed model of LAT-1 function. LAT-1 interacts with LAG-2 on the DTC to boost GLP-1 activation in germ cells. This interaction leads to the modulation of *gld-1* expression and subsequently, an increase in germ cell proliferation and the regulation of progenitor zone size. Red arrows and lines highlight steps investigated in this study. Graph raw data are provided in the Source Data. Graph details and statistics are: (**a**): Mean ± SD. (**c**), (**d**), (**e**): Box plots with median (center), interquartal range, 5th (lower whisker) and 95th (upper whisker) percentiles. One-way ANOVA with Bonferroni post-hoc test.

suppress the mRNAs of downstream targets like *gld-1*[64] (Fig. 2a). Consequently, the reduced Notch activation observed in *lat-1* mutants likely accounts for the detected increase in GLD-1 protein levels and its presence in more distal germ cells (Fig. 2b, c), a phenotype similar to that in *glp-1* reduction-of-function mutants (allele *bn18*)[65].

Through this signaling mechanism, Notch activity regulates the balance between cell proliferation and differentiation into germ cell progenitors (i.e., entry into meiosis) in the distal germline of *C. elegans* (summarized in ref. 26). As germ cells move away from the DTC and thus, from the ligand LAG-2 located on this cell, Notch activity declines, leading to meiotic differentiation (summarized in ref. 26). Worms with reduced GLP-1 function exhibit a smaller progenitor zone and fewer proliferating cells[25,27], a phenotype also observed in the absence of LAT-1 (Fig. 1).

Beyond Notch signaling, there are indications that LAT-1 may also play roles in cell cycle regulation in the germline. This is suggested by the altered M and S phase indices in *lat-1* mutants, implying a faster cell cycle. While altered Notch signaling does not typically affect the mitotic index[25,32], a cross-talk between Notch and cell cycle regulation components has been reported[66,67]. *lat-1* expression in both the DTC and germ cells hints at further roles that warrant investigation.

Notably, although Notch activity is required throughout the reproductive lifespan, time-course analyses indicate that LAT-1 plays a specific role in young adult nematodes (8–12 h post-L4 stage) (Supplementary Fig. 1a), when their reproductive capacity starts to build up[27]. This suggests that LAT-1 enhances Notch signaling during this critical period, interacting with LAG-2, one of two conserved DSL homologs in *C. elegans*[21,68].

In multiple distinct lines of evidence, we show the interaction between LAT-1 and LAG-2 through their extracellular regions. Our AlphaFold2 Multimer-generated models suggest that both the LAT-1 RBL and GAIN domains form the interaction interface with the GAIN domain contributing the larger area. These domains have been previously shown to be crucial for both receptor 7TM-dependent and 7TM-independent/N terminus-only/*trans* functions[62]. Further support for this interaction is provided by BRET and BiFC analyses in both cells and live nematodes identifying the key amino acids involved (Figs. 3, 4). It needs to be noted that a larger interface does not necessarily imply a greater functional contribution and/or physiological effect. Indeed, our mutational analyses (Fig. 3c) reveal that only the combined mutations of both RBL and GAIN domains significantly impairs binding affinity, whereas mutations in either domain alone have little to no effect. This suggest that the two domains function cooperatively to establish a stable and specific interaction.

Further investigation into additional proteins potentially involved in the LAT-1/LAG-2 interaction could provide a deeper understanding, as latrophilin homologs in mice neurons function within large protein super-complexes[69]. For instance, while we could not detect a direct interaction between LAT-1 and GLP-1, it remains possible that LAT-1

forms a complex with LAG-2 and GLP-1 when all three proteins are present.

We provide physiological and mechanistic evidence that only the N terminus of LAT-1 is sufficient to modulate Notch activity indicating that the full-length receptor is not required for this function. Reduced Notch signaling was ameliorated by expression of the *N* terminus tethered to the membrane (LNT), indicating that the LAT-1 function is indeed able to modulate Notch pathway activity in a non-cell autonomous manner. This is consistent with previous findings showing that many functions of the aGPCR in the germline, such as sperm guidance and germ cell apoptosis, rely solely on its *N* terminus[22]. These data raised the question of whether the interaction between LAT-1 and LAG-2 occurs *in cis* (on the same cell) or in trans (on different cells). Both in vitro and in vivo analyses (Fig. 5) support a *cis* interaction, implying that LAT-1 binds to LAG-2 on the same cell and that the *N* terminus-only/7TM-independent function of the aGPCR is based on a *cis* interaction in the DTC, but it exerts an effect in trans on neighboring germ cells.

In summary, our results propose a model in which LAT-1, via its *N* terminus, interacts with LAG-2 on the somatic DTC, directly modulating the Notch ligand-receptor complex to regulate the expansion of the germline stem cell pool in *C. elegans* (Fig. 5f). This role might complement the previously identified functions of LAT-1 in regulating brood size, as the latter was partly rescued by LNT expression in the DTC[22]. Interestingly, our data indicate that LAT-1 may also engage in cross-talk with the Notch pathway in additional contexts, such as anus development and chemosensory regulation (Fig. 2), potentially pointing to a general underlying mechanism.

The question remains how the interaction of latrophilin with the Notch ligand promotes increased Notch activation. Several modulators of the Notch pathway have been identified in the past, acting at different sites, for example regulating expression of key Notch components[70] or influencing their degradation via ubiquitination[71]. LAT-1 does not alter the overall expression of *lag-2* or *glp-1* (Supplementary Fig. 2). However, it may stabilize LAG-2 on the membrane, elevating signal intensity. Alternatively, LAT-1 could increase the effect of LAG-2 on GLP-1. It is well established that for the activation of the Notch pathway, the Notch ligand binds to the Notch receptor on opposing cells (summarized in refs. 58,72). Subsequently, a certain level of force exerted from the Notch ligand is required to activate the receptor, which leads to the exposure of protease recognition sites and receptor cleavage at two sites, one extracellularly and one in the transmembrane domain that releases the NICD (summarized in refs. 58,72). In *Drosophila melanogaster* and vertebrates, this mechanical force needs to be quite strong and is generated by ligand endocytosis, promoting a conformational change within the Notch receptor. In *C.elegans*, much lower force thresholds are in place, tethering the ligand to the membrane is sufficient[73]. A hypothesis could be that LAT-1 acts as an allosteric modulator, enhancing the conformational changes in the LAG-2/GLP-1 complex required for signaling.

This newly discovered mechanism of LAT-1 interacting with LAG-2 could be generalizable and may also be present in other contexts within *C. elegans* or even in other species. Our analyses hint towards a broader involvement of the two at least in the nematode, since in anus development and neurological contexts such as octanol avoidance an interaction is conceivable (Fig. 2). In addition, a cross-talk could occur in embryonic development. Notch signals are required for specifying cell fates in the embryo and for mediating correct embryonic axis formation[45–47] with LAG-2 acting as a key signal in certain embryonic cells[74,75]. As LAT-1 is also present in the early embryo[36], an interaction in this context is plausible.

In mammals, none of the three latrophilin homologs has been directly linked to Notch signaling. However, a cross-talk between aGPCRs and Notch components has been hypothesized. For example, overexpression of human *ADGRL4/ELTD1* leads to a downregulation of the Notch ligand *DLL4* and an upregulation of ligand *JAG1*[76]. Conversely, it was shown in mice that expression of *DLL4* increased *ADGRG6/GPR126*[77] expression. However, a direct interlink was lacking to date. Our work uncovers first mechanistic insights on how an aGPCR interacts with the Notch pathway in a multicellular setting in vivo. This is particularly relevant given the essential role of Notch in development, tissue homeostasis, and pathologies such as cancer[52–55], which makes new potential regulators of the Notch pathway highly relevant for pharmacological and medical research.

## Methods

### Materials and reagents
All standard chemicals were from Sigma Aldrich, ThermoFisher Scientific or Carl Roth unless stated otherwise. All enzymes were obtained from New England Biolabs unless stated otherwise. Details are given in Supplementary Table 1.

### Generation of constructs and plasmids
Constructs were generated using standard restriction-ligation or HiFi assembly (New England Biolabs) (see Supplementary Methods for details). Oligonucleotide sequences are given in Supplementary Table 2 and generated constructs in Supplementary Table 3.

### *C. elegans* maintenance and strains
*C. elegans* strains were maintained according to standard protocols[78] on *E. coli* OP50 at 15 °C unless stated otherwise. A complete list of used strains is given in Supplementary Table 4. Assays were performed with worms at L4 + 8 h kept at 15 °C unless stated otherwise.

### Generation of transgenic *C. elegans* lines
For BiFC experiments, transgenic strains with stably transmitting extrachromosomal arrays were generated employing standard injection techniques into *C. elegans* N2[79]. Nematodes were injected with BiFC plasmids (Supplementary Table 4). Co-injection markers were either pRF4[80] (100 ng/µL), pCFJ90 (2.5 ng/µL)[81] (gift from Erik Jorgensen, Addgene plasmid #19327), pCFJ104 (5 ng/µL)[81] (gift from Erik Jorgensen, Addgene plasmid #19328), or IR98 (30 ng/µL)[82]. pBluescript II SK+ vector DNA (Stratagene) was added as stuffer DNA to achieve a final concentration of 120 ng/µl.

For competition BiFC, plasmid pSP234 was injected in various concentrations (5, 10, 50 ng/µL) into N2 nematodes with IR98 (Hygromycin resistance) as co-injection marker, and crossed into respective BiFC lines, thereby retaining its original extrachromosomal array composition. Transgenic progeny was isolated and stably transmitting lines were selected.

For rescue experiments, strains were generated by CRISPR-Cas9 genome editing. This was accomplished using either a Cas9-RNP complex[83–86], or via a plasmid-based approach[87–90]. For detailed descriptions of modifications see Supplementary Methods, for a list of genetically modified/transgenic strains see Supplementary Table 4.

### Antibody and DAPI staining
Antibody and DAPI stainings were performed on extruded germlines. For this purpose, germlines were dissected in PBS and fixed. For anti-phospho histone H3 (anti-PH3) staining, fixation was performed in 3.7% (v/v) formaldehyde in PBS containing 0,1% (v/v) Tween 20 (PBS-T). Gonads were post-fixed in methanol at −20 °C for 5 min. No blocking was required. For staining the NICD::V5, which followed the protocol described in ref. 34, gonads were fixed with 4% (w/v) paraformaldehyde for 10 min. Permeabilization was performed using 0.1% (v/v) Triton X-100 in PBS (PBS-T) for 30 min followed by blocking in 0.5% (w/v) BSA in PBS-T for 20 min. Intermittent washing was done thrice using PBS-T for 5 min per wash. All centrifugation steps were performed at 3000 x *g* for 1 min.

Thereafter, gonads were incubated with the primary antibody overnight at 4 °C (anti-PH3: rabbit anti-phospho histone H3 (Ser10)) 1:200 in PBS-T/0.1% BSA; anti-V5: mouse anti-V5 SV5-Pk1 1:1000 in PBS-T/0.5% BSA) while rotating. After washing three times in PBS-T, gonads were incubated with the secondary antibody (anti-PH3: goat anti-rabbit IRDye 680RD-conjugated 1:1000 in PBS-T/0.1% BSA supplemented with 10 ng/µL DAPI; anti-V5: goat anti-mouse IgG (H + L), F(ab')2 fragment CF 568 1:1000 in PBS-T/0.5% BSA supplemented with 10 ng/µL DAPI) at room temperature for 1 h. Gonads were mounted in Fluoromount G mounting media on 2% agarose pads. For details regarding antibodies or reagents, please refer to Supplementary Table 1.

### EdU labeling
5-ethynyl-2′-deoxyuridine (EdU) staining of worms was conducted as previously described[91,92]. For incorporation of EdU into bacteria, an overnight culture of MG1693 bacteria (*E. coli* genetic stock center) was diluted 1:50 in M9 containing 1% glucose, 1.25 µg/mL thiamine, 0.5 µM thymidine, 1 mM $MgSO_4$ and 20 µM EdU (Click-iT EdU Alexa Fluor 647 Imaging Kit, ThermoFisher) and grown for 24 h at 37 °C. Staged adult hermaphrodites (8 h post L4) were transferred to NGM plates seeded with these bacteria and incubated for 30 min in the dark at room temperature. Subsequently, worms were washed off the plates and gonads were extruded and fixed as described for the PH3 staining. EdU staining was completed using the Click-iT EdU Alexa Fluor 488 or 647 Imaging Kit (ThermoFisher) according to the manufacturer's protocol.

### Microscopy
Microscopy of nematodes was performed using a Leica TCS SP8 confocal microscope equipped with LAS X software. Z stacks were taken with spatial spacing of 0.5–2 µm, depending on the specimen (2 µm for whole worms, 0.5 µm for gonads) using either 40x or 93x magnification objectives. Microscopic images were evaluated using Fiji[93]. Scoring of anus phenotypes was performed using a Zeiss AXIO Imager.A2 using 63x magnification and DIC.

### Notch activation assay
GLP-1 activation was monitored in worms carrying the allele *glp-1(q1000[glp-1::4xV5])* that has a 4x V5 tag located in the GLP-1 intracellular domain (NICD)[34]. These were stained with an anti-V5 antibody as described above, mounted, and subjected to microscopy. Fiji[93] was used to quantify Notch activation as the fraction of NICD present in nuclei in a central *Z* slice. For this purpose, a central slice was selected, and nuclei in the first five rows were annotated as ROI based on DAPI staining. The fluorescent signal of the V5 staining in these areas was quantified. NICD ratio was calculated as: sum [V5 signal in DAPI-annotated ROIs] / total V5 signal in slice.

### Assessing DTC morphology
The DTC morphology was quantified in worms carrying allele *qIs153[lag-2p*::MYR::GFP + *ttx-3p*::DsRed] V, which encodes a myristylated GFP marking the DTC membrane. Gonads were extruded,

fixed, stained with DAPI, mounted, and imaged using confocal microscopy as described above. Quantification of DTC structures was performed according to ref. 94. In brief, the DTC cap was defined as from distal to the most proximal extent of signal covering the germ cell surface, whereas the plexus was defending from the most distal signal to the most proximal signal of intercalating processes.

## Quantification of GLD-1 levels
Gonads of 8 h post-L4 worms containing *ozIs5[gld-1::gfp + unc-119( +)]*[95] were extruded, fixed, stained with DAPI, and mounted as described above. To compare expression in different strains images were acquired with same exposure and detection settings. Z projections of the image stacks were analyzed as previously described[20] with the "Plot Profile" function in Fiji[93] to generate expression profiles of the distal germ cell rows (ROI: a 5 μm-wide square spanning 35 germ cell rows). Fluorescence intensities of each germ cell row were obtained by dividing the plot profile into 35 equal sections, respectively. For each germ cell row, a mean fluorescence value was calculated by pooling fluorescence intensities of multiple germlines.

## Anus absence quantification
Adult nematodes were bleach-synchronized as per standard procedure and hatched overnight at room temperature. L1-arrested nematodes were immobilized using 300 mM levamisole in M9 and mounted on 2% agarose pads. Nematodes were subjected to microscopy, and the absence of the anus was scored based on tail morphology.

## Octanol avoidance assay
Nematodes were raised at 25 °C (restrictive temperature for *lag-2* and *lat-1; lag-2* mutants). At 1-day-old adult stage, they were assessed for their chemosensory response as described in ref. 96. Briefly, freshly diluted octanol (70% in ethanol) was soaked into an eyelash and placed in front of a forward-moving animal on an NGM agar plate without food. As a negative control, 100% ethanol was used. The time required to initiate reversing was recorded.

## BiFC analysis
Adult nematodes expressing BiFC constructs were heat-shocked for 3 h at 33 °C and recovered for 5 h at 22 °C prior to anesthetizing in 300 μM levamisole in M9.

Z stacks of nematode heads were acquired using confocal microscopy using 2 μm spacing. Images were evaluated using Fiji[93]. Quantification was performed using the "sum slices" function and quantifying the fluorescence using the anterior end of the head until the posterior end of the pharynx. Gut autofluorescence signal recorded from non-heat-shocked nematodes was subtracted to compensate for background signal.

## Western blot analysis
60-80 1-day-old adult nematodes were heat-shocked for 3 h at 33 °C, recovered for 5 h at 22 °C, and snap frozen at −80 °C in 20 μL M9. Samples were boiled for 10 min at 95 °C in 1x Laemmli buffer and spun down. 20 μL of supernatant were subjected to PAGE on a 10% SDS polyacrylamide gel and transferred on a nitrocellulose membrane. After blocking with EveryBlot buffer (BioRad) for 5 min at room temperature, the membrane was incubated with a mix of the two primary antibodies mouse anti-V5 SV5-Pk1 (BioRad, MCA1360) and rabbit anti-HA (Abcam, ab236632) (each 1:1000 in EveryBlot buffer) overnight at 4 °C. Unbound antibody was washed off using TBS-T (0.1%) in three washes. Secondary antibody incubation was performed using the anti-rabbit IgG, HRP-conjugated antibody (Cell Signaling Technology, 7074) (1:2500 in EveryBlot buffer) and the goat anti-mouse IgG (H + L)-HRP conjugate (BioRad, 1721011) (1:5000 in EveryBlot buffer) with anti-actin hFAB-rhodamine (Biorad, 12004164) (1:2500 in EveryBlot buffer) as loading control. Incubation with the secondary antibodies was

performed for 1 h at room temperature followed by washing as stated above, followed by detection of HRP-conjugated antibodies using SuperSignal West Pico PLUS (Thermo Fisher, 34577) as well as fluorescent detection of actin by exciting at 519 nm and detecting at 605 nm using a UVP Chemstudio (Analytik Jena).

## Molecular modeling of LAT-1 and LAG-2
The models of the extracellular regions of LAT-1 and LAG-2 were constructed using AlphaFold2 Multimer[39,97]. The software was downloaded from Github (https://github.com/deepmind/AlphaFold) and installed on the local computing cluster of Leipzig University Rechenzentrum (10/2022, v2.2.2). Sequences of the extracellular domains and combinations thereof were obtained from Uniprot[98] and used as input for various complex combinations of LAG-2 and LAT-1. Here, models of the entire extracellular region of LAG-2 with a) only the GAIN domain, b) GAIN and HRM domain, c) only the RBL domain, d) LAT-1 with an extended linker sequence between RBL domain and GAIN domain and e) the entire extracellular region of LAT-1 were predicted. Input multiple sequence alignment (MSA) features were generated by the AlphaFold2 Multimer pipeline described in ref. 97. For each interaction combination, 50 models were generated, energetically minimized, and ranked. Obtained structures were analyzed and visually inspected in PyMOL (version 2.5.4). Initial root-mean square deviation RMSD calculations were conducted using PyMOL alignment calculation. Each interaction complex combination was analyzed, and common interaction faces inspected. The most promising models, defined by the highest ipTM and pTM as AlphaFold2 scores and with the most commonly observed interactions, were energetically minimized (Rosetta relax) using Rosetta3 (version 3.13)[99]. For each model, 300 minimized structures were generated, and the interface was analyzed. Here, the Rosetta InterfaceAnalzyer was used, and the top-scoring model was selected for further analysis. The quality of the structural model and its interaction were assessed with MD simulations.

To study the energetics of the interaction, a per-residue energy analysis (energy breakdown) was carried out using Rosetta's per_residue_energies application. Based on the individual energies for residue-residue interactions between the different protein domains, a residue contact map (hotspot analysis) was made and plotted with Python (version 3.7). Based on a hotspot analysis and after visual inspection using PyMOL, relevant interactions were analyzed by mutagenesis experiments. In an additional refinement step, the investigated mutations were modeled with AlphaFold2 Multimer and compared to the initial models. The structural model of the mutations was analyzed similarly. A protocol capture can be found in the Supplementary Methods.

To investigate the complex of LAT-1 and GLP-1, the aforementioned computational pipeline was utilized, but in both cases, the predicted complexes had lower in silico scores and lacked a convergence with highlight variable binding interfaces.

## MD simulations
The best-scoring complex model containing LAG-2 and the RBL and GAIN domains of LAT-1 was used as the basis for the MD system. Disordered non-contacting regions were truncated, leaving residues LAT-1 32-137 (RBL) and 182-539 (GAIN) as well as residues 16-273 for LAG-2 in the system. The termini were capped via N-terminal amidation and C-terminal acetylation except for the N-terminus of LAG-2, putatively C-terminal of signal peptide cleavage, which was left as a charged amino group. Disulfide bridges were added according to the respective UniProtKB database entries (*lag-2*: P45442; *lat-1*: G5EDW2)[100]. The truncated complex was processed for residue hetero states and side chain flips by maestro 2024.2. CHARMM-GUI[101,102] was used to normalize bond lengths and generate minimization and equilibration inputs using the CHARMM36 forcefield for GROMACS (version 2024.2). Water boxes

using the TIP3P water model[103] were generated with CHARMM-GUI using a large cubic box size of 150 Å edge length to accommodate expected large-scale rigid body movements and charge-neutralized with 0.15 M NaCl. After minimization using the steepest-descent method for 5000 steps, a 50 ns equilibration with 1 fs time step was performed with backbone restraints of 400 kJ/mol/nm² and side-chain restraints of 40 kJ/mol/nm² to yield the equilibrated model and extensively accommodate equilibration of the predicted structure. After equilibration, triplicate unbiased MD simulations for 1000 ns using the CHARMM36 force field[104] in GROMACS[105,106] were performed at 295 K using the c-rescale barostat, the v-rescale thermostat and a 2 fs time step. MD analysis was carried out with GROMACS 2024.2 within a python3 environment and by using VMD[107]. For a summary of data reliability and reproducibility, see Supplementary Table 5.

## Cell culture
HEK293 and HEK293T cells were maintained in Dulbecco's minimum essential medium (DMEM, high glucose, + L-Glutamine)/F12 (1:1, v/v) supplemented with 15% heat inactivated fetal bovine serum (FBS) in humidified atmosphere at 37 °C and 5% $CO_2$.

## Cell surface ELISA
Cell surface expression of receptors carrying an N-terminal HA tag was determined by an indirect enzyme-linked immunosorbent assay (ELISA). HEK293T cells were split into 48-well plates ($1.2 \times 10^5$ cells/well) that had been pre-coated with 0.0002% poly-L-lysine (Merck) in PBS (30 min at 37 °C). After 24 h, the cells were transfected with 500 ng of receptor-encoding plasmid DNA per well using Lipofectamine 2000 (ThermoFisher), following the manufacturer's instructions. Cells were fixed with 4% formaldehyde for 20 min and blocked with media + 10% FBS for 1 h at 37 °C. After washing with PBS once cells were incubated with an anti-HA peroxidase conjugated antibody (1:1000) (Sigma-Aldrich, 54193500) and detection was performed as previously described[108]. Briefly, after extensive washing, $H_2O_2$ and o-phenylenediamine were added to the wells (2.5 mM each in 0.1 M phosphate-citrate buffer (pH 5.0)). The color debelopment was stopped after 15 min by addition of 1 M $H_2SO_4$ and cell surface expression was measured at 492 and 620 nm in a Spark plate reader (Tecan).

## BRET analyses
HEK293 cells were seeded into 6-well plates ($1.25 \times 10^6$ cells/well) and transiently transfected with a DNA gradient of *Venus::lat-1* (or its variants; 0-3970 ng DNA/well, six conditions) over a constant amount of *Nluc::lag-2* (30 ng DNA/well) using 3 µl MetafectenePro (Biontex)/ µg DNA (for BRET plasmids see Supplementary Table 3). The total DNA per well was 4000 ng and was balanced out by empty vector. 16 h post transfection, the cells were detached and re-seeded into poly-D-lysine-coated solid white 96-well plates (BRET reading) or clear bottom black 96-well plates (direct fluorescence excitation) (150.000 cells/well) using phenol-red free medium. One day later, the medium was changed to 200 µl Hank's balanced salt solution ( + $Ca^{2+}$/$Mg^{2+}$) supplemented with 25 mM HEPES (pH 7.4; BRET-buffer). Protein expression of *Venus::lat-1* and *Venus::hGLP1R* was quantified in the black plates by direct excitation at 485 ± 20 nm, and fluorescence emission was recorded at 544 ± 25 nm ($F_{ex}$). For the BRET reading, the luciferase substrate coelenterazine H (Nanolight Prolume) was freshly added into the BRET buffer (final concentration 5 µM) and the cells were incubated for 5 min at 37 °C before measuring dual-color luminescence with filters 400–470 nm (L) and 535–650 nm (F) in a Spark plate reader (Tecan) using the well-wise mode and 500 ms signal integration time.

BRET ratios were calculated as (F/L) and subsequently corrected by the BRET ratio observed in donor-only cells (0 ng *Venus::lat-1* variant) to retrieve the netBRET. For any *Venus::lat-1* variant, the change in netBRET was plotted against the protein expression ratio of *Venus::lat-*

*1* over *Nluc::lag-2* luminescence in the donor-only state ($F_{ex}$ /$L_0$). Results from four independent experiments were combined and fit to a one site total binding hyperbola function (Y=Bmax*X/Kd+X) + NS*X) using GraphPadPrism 10, where Y is the netBRET and X is $F_{ex}/L_0$, treating the nonspecific component of the binding hyperbola (NS) as a shared optimization parameter due to random collision. Differences in $K_d$ (equivalent to $BRET_{50}$) reflect changed affinity of the specific binding between LAG-2 or APX-1 and the respective LAT-1 variant.

To evaluate whether the proteins interact in trans, i.e., between neighboring cells rather than on the same cell, two distinct HEK293 cell populations were transfected with either *Nluc-lag-2* (200 ng plasmid DNA/T25 flask) or *Venus-lat-1* (4000 ng/T25 flask). One day after transfection, the cells were re-seeded into poly-D-lysine coated white and black 96-well plates in technical quadruplicate using phenol red-free medium, shaping a gradient by increasing the cell number of *Venus::lat-1*-transfected cells and balancing the total number of cells by filling up with untransfected HEK293 (150.000 cells/well in total; 3 cell populations). In an alternative approach, different amounts of *Venus::lat-1* DNA were transfected (0–4000 ng DNA/well in 6-well-plates), and the fluorescent gradient was shaped by using 130.000 cells of these different *Venus::lat-1* expressing cell populations, combined with 20.000 *Nluc::lag-2* expressing cells (150.000 cells/well in total; 2 cell populations). 24 h after re-seeding, the fluorescent gradient was confirmed by direct excitation in black plates and the BRET was measured as described above.

## Statistics
Statistical and graphical analyses (except for the molecular modeling data) were performed using Prism version 10.0 (GraphPad Software). When comparing two groups, statistical significance was analyzed using the two-sided unpaired Student's t test. When comparing several groups, a one-way ANOVA was applied with Bonferroni post-hoc tests to correct for multiple comparisons unless stated otherwise. Data are presented as box plots unless stated otherwise. Details are given in the figure legends.

## Reporting summary
Further information on research design is available in the Nature Portfolio Reporting Summary linked to this article.

# Data availability
Source data and supplementary information are provided with this paper. All other data and worm strains generated in this study are available upon request to the corresponding author.

# Code availability
Data from in silico predictions and MD simulations are stored in Zenodo repository under https://doi.org/10.5281/zenodo.15228753. Source data are provided with this paper.

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

## Acknowledgements

We thank Franziska Fiedler, Niko Fleischer, and Anna Tisnikar for help with generating constructs and establishing phenotyping assays, Hannah Mönch and Daniel Fox for support with data analysis, and Claudia Binder, Diane Schmiegelt, Barbara Klüver, and Ronald Weichelt for technical assistance. We are very grateful to Mike Boxem, John Calarco, Oliver Hobert, Erik Jorgensen, Ralf Schnabel, and Allison Woollard for kindly providing plasmids, W. Brent Derry and the *Caenorhabditis Genetics Center* (CGC) (which is funded by the NIH Office of Research Infrastructure Programs (P40 OD010440)) for generously sharing *C. elegans* strains. We would further like to acknowledge the Center for Advanced Imaging (CAi) at the Heinrich Heine University Düsseldorf for providing access to the Leica TCS SP8 STED 3X and the Abberior Facility Line, and especially Sebastian Hänsch for general support with imaging and analysis. This work was supported by a scholarship to D.M. from the Medical Faculty, Leipzig University, and by a Humboldt Professorship to J.M. from the Alexander von Humboldt Foundation. The authors acknowledge grants from the Deutsche Forschungsgemeinschaft (DFG, German Research Foundation) through CRC 1423/2 (project number 421152132; B03 (A.K.), C04 (S.P., T.S.), Z04 (P.W.H., J.M.)), FOR 2149 (project number 246212759; P02 (S.P.) and P04 (T.S.)), and SPP2363 (project number 460865652 (J.M.)). Funding for instrumentation: Leica TCS SP8 STED 3X: DFG INST 208/665-1 FUGG; Abberior Facility Line: DFG INST 208/805-1 FUGG. Financial support was also provided by the Federal Ministry of Education and Research of Germany and by the Sächsische Staatsministerium für Wissenschaft, Kultur und Tourismus in the program Center of Excellence for AI Research, Center for Scalable Data Analytics and Artificial Intelligence Dresden/Leipzig (project identification number SCADS24B (F.L., J.M.)).

## Author contributions

S.P. conceived and designed the study. W.B.P.: cloning, transgene generation, genetic modification, worm genetics, PH3 assays, anus phenotyping, Notch activation assays, Western Blot, BiFC assays, and data analysis. V.E.G.: worm genetics, DAPI, PH3, REC-8, EdU, and GLD-1 assays, data analysis. D.M.: worm genetics, design of DAPI, PH3, REC-8, EdU, GLD-1, and DTC morphology assays, data analysis. I.C.: ELISA experiments, octanol reversion assays. F.L. and J.M.: molecular modeling. F.S. and P.W.H.: MD simulations. A.K.: BRET assays. T.S.: data analysis. W.B.P., V.E.G., D.M., and S.P. wrote the manuscript with the consent of all co-authors. All authors agreed to the final version of the manuscript.

## Funding

## Competing interests

The authors declare no competing interests.
