## [Transparent Peer Review file · Nature Communications]

Notch activity is modulated by the aGPCR Latrophilin binding the DSL ligand in *C. elegans*

Corresponding Author: Professor Simone Prömel

Version 0:

Reviewer comments:

Reviewer #1

(Remarks to the Author)

Post et al. explore the aGPCR latrophilin with respect to Notch signaling, proposing that it increases activation of the Notch receptor by cis-cell interaction with a ligand for Notch through their extracellular domains. The structure function analysis of the proposed protein-protein interaction is convincing. However, the germline readout is modest and there are concerns regarding the interpretation of these results. A biological readout for a possible interaction between Notch signaling and LAT-1 requires more support.

Comments

1. Figure 1. Loss of function of many genes in the genome can cause modest effects on the germline progenitor zone, and lat-1 falls in this group. The mitotic index seems very similar (panel h) despite the reported statistical difference. lat-1 modestly affects the overall accumulation of germ cells, but the claim that "LAT-1 plays a role in controlling cell proliferation and/or regulating zone size during the late L4/very early adult stage" would require a time-course analysis.
2. Figure 2. The claim that lat-1 is affecting GLP-1 signaling is not well supported. While a double mutant of two null alleles with no additive effect over each single mutant would suggest a possible linear pathway, that is not the case presented here and so the interpretation of the data presented in Figure 2 is problematic. GLP-1 Notch is thought to function in cell fate (stem versus differentiated), but not in proliferation rate. Typically, genes that act in the GLP-1 pathway cause premature differentiation (temporal or spatial) such that their double mutant combinations with a partial loss-of-function glp-1 allele results in complete loss of the entire progenitor zone (e.g., cye-1). Conversely, genes that act in proliferation (e.g., daf-2, eat-2) cause changes in the mitotic profile (e.g., M-phase index) but not the same "glp-1-like phenotype" (loss of all progenitor cells). In these cases, the interpretation is that strong enhancement of the "glp-1-like phenotype" in the sensitized low-Notch signaling background is evidence for involvement in the pathway whereas a minor reduction in progenitor number is not.
3. Further support for the interaction with lag-2 could be demonstrated with other lag-2 phenotypes, such as the many interactions between lag-2 and lin-12.
4. Does LAT-1 also bind DTC-expressed APX-1 (at least in silico?), or generally to proteins with EGF-like extracellular motifs?
5. Figure 4. Demonstration of LAG-2 and LAT-1 co-localization on the DTC membrane would provide additional support. It is unclear why the assay was confined to the head.
6. Figure 5. The modest effect on the germ line and the high level of variability in the data make it difficult to interpret the locus of action. In addition, the mitotic index, rather than number of PH3 positive cells, should be reported given that the strains vary in number of progenitor cells.

Reviewer #2

(Remarks to the Author)

Post et al. identify the Adhesion GPCR latrophilin (LAT-1) as a positive modulator of the conserved Notch signaling pathway, which regulates critical processes like cell proliferation, differentiation, and tissue homeostasis. Through in silico, in vitro, and in vivo analyses, the authors reveal that LAT-1 directly interacts with the Notch ligand LAG-2 via its conserved GAIN and RBL domains. This modulation occurs independently of LAT-1's 7TM region and G protein signaling, uncovering a unique mechanism of action for aGPCRs in fine-tuning Notch signaling.

Comments:

1. A more complete scheme on the Notch signaling pathway (like the one shown in <https://www.sciencedirect.com/science/article/pii/S0092867409003821?via=ihub#fig1>) for the general readership of Nat. Com. would be useful to better understand the biological context of this work. The one in Figure 2a is insufficient, as it does not show important elements discussed in this study such as NCID, Notch receptor (not explicitly mentioned) or the LAT-1 and its discussed effect.
2. The authors predicted interactions between the extracellular domains of LAT-1 with LAG-2 and GLP-1, using AlphaFold2 Multimer. I find this structural model very intriguing but it also leaves many open questions. Did the authors investigate the model quality/stability by carrying out MD simulations?
3. The binding interface of the GAIN domain seems to be larger compared to the RBL domain. Would this suggest that the GAIN domain contributes more to the binding?
4. Overall the reader would appreciate more information about binding interfaces, including interface size, amount of polar and hydrophobic interactions, etc.
5. The authors analyze alanine mutations of the RBL and GAIN domain. For instance, they find that the RBL or GAIN mutant alone does not significantly reduce the affinity of LAT-1 for LAG-2(Figure 3c). Would this imply that one domain alone is sufficient to mediate the observed binding including the in vivo function? Can the authors carry out more experiments to investigate this possibility?
6. In Figure 3a, the LAG-2 structural representation should be explicitly labeled as such to enhance clarity, and each element in the structure should be described in the caption (e.g.: "GAIN domain (yellow cartoon)").
7. Extended Data Figure 4 should add a label for the vertical left residues in both plots and explain what is in the top and bottom. In addition, the cell and x-axis numbers are quite small. Color scale range [0 to 1] does not match the numeric ranges of the figures [-71.1 to 1.7]
8. Line 182 add Fig 3b reference.
9. Extended Data Figure 5 uses Rosetta Energy Units (REU). Please provide more information on this unit.
10. Figure 3 should be better described in the caption, for example adding the respective explanation with the coloring for the structures (Yellow, RBL domain, Grey surface LAG-2...etc.)
11. HBD loop. the HBD abbreviation in Fig 3, is not explained. Also, this loop seems to be named differentially in "Supplementary Protocol Capture Molecular Modeling" as HRM, HormR, and HBD (?) in Figure 3. Please check.
12. The Method section "Molecular modeling of LAT-1 and LAG-2" and the "Supplementary Protocol Capture Molecular Modeling" are not clearly explained and should be revised. For example, the order of the process changes in both explanations. A flowchart or figure would enhance the understanding of this section. Please revise.
13. The authors mention: "While no interaction was predicted between LAT-1 and GLP-1, an interaction between LAT-1 and LAG-2 appeared likely, with the interaction likely occurring via the LAT-1 Rhamnose-binding lectin (RBL) and GPCR autoproteolysis-inducing (GAIN) domains (Fig. 3a, Extended Data Fig. 4)". Here, the word "likely" should be revised. Use better scores of the provided AlphaFold model.

Version 1:

Reviewer comments:

Reviewer #1

(Remarks to the Author)

In general, the authors have addressed this reviewer's major concerns. However, there are still several potentially misleading that need to be addressed:

Line 109:

"a decreased number of proliferating cells and an altered proliferative zone size have been observed in glp-1 loss-of-function

mutants 16,25”

First, there are no proliferating germ cells in “glp-1 loss-of-function” mutants. The authors are referring to reduction-of-function scenarios in temperature sensitive (ts) mutants. Second, the “proliferative zone” was renamed within the field over the last several years as “progenitor zone” to conform with the wider stem cell biology field. This term should be replaced throughout the manuscript and figures.

Line 129

“Similar to lat-1 mutants, hermaphrodites carrying single loss-of-function mutations in lag-2 (lag-2(q420) 31) and glp-1 (glp-1(bn18) 32) exhibit defects in germ cell proliferation and meiotic entry...”

And Line 329

“phenotype similar to that of glp-1 loss-of-function mutants (allele bn48)”

Neither lag-2(q420) nor glp-1 (glp-1(bn18) are “loss-of-function” alleles. They are both ts reduction-of-function alleles. This should be clarified. And, in 329, the allele is presumably bn18?

Line 391

“However, only subsequent ligand endocytosis truly activates the receptor by generating a mechanical force that promotes a conformational change within the Notch receptor.”

Mechanical force on the DSL ligand is not thought to be a required mechanism for Notch receptor activation in *C. elegans* (see Langridge et al. 2022)

Reviewer #2

(Remarks to the Author)

The authors have addressed all of my comments satisfactorily and have substantially improved the manuscript in the revised version. I have no further concerns.

Notch activity is modulated by the Adhesion GPCR Latrophilin binding to the DSL ligand in the germline stem cell niche in *C. elegans*

Response to Reviewers' comments

We thank the reviewers for the overall positive feedback and the helpful comments on our manuscript. In response to their suggestions, we have performed additional experiments to further support and strengthen our findings. Moreover, we have incorporated their constructive input to improve the clarity and overall impact of the study.

Below, we address all reviewer comments in detail and quote statements from the reviewers in **bold face**.

Reviewer #1

Major comments

- 1. I) Figure 1. Loss of function of many genes in the genome can cause modest effects on the germline progenitor zone, and *lat-1* falls in this group. The mitotic index seems very similar (panel h) despite the reported statistical difference.**

We agree with the reviewer that the effects of LAT-1 in the germline are modest. However, they are both significant and reproducible. To further strengthen these data, we have performed additional experiments (e.g., Fig. 5c–e). In the first version of the manuscript, the sample size was $n \geq 25$ across at least 6 independent experiments; in the revised version, it is now $n \geq 31$ across at least 7 independent experiments. These additions have resulted in a more robust and reliable data set.

II) *lat-1* modestly affects the overall accumulation of germ cells, but the claim that “LAT-1 plays a role in controlling cell proliferation and/or regulating zone size during the late L4/very early adult stage” would require a time-course analysis.

We apologize if the original sentence was misleading. We have included a time-course analysis examining the effects of LAT-1 loss on both the total number of germ cells and the number of PH3-positive germ cells, as well as the corresponding mitotic indices, in hermaphrodites from the L4 stage up to 72 h post-L4 (Supplementary Fig. 1a-c). To better highlight these findings, we have now added a direct reference to these data in the Results section: *“Time-course analyses revealed that these phenotypes were most prominent in worms aged L4 + 8 h (Supplementary Fig. 1a-c).”*

Furthermore, we have added an explanatory sentence in the Discussion: *“Notably, although Notch activity is required throughout the reproductive lifespan, time-course analyses indicate that LAT-1 plays a specific role in young adult nematodes (8-12 hours post-L4 stage) (Supplementary Fig. 1a), when their reproductive capacity starts to build up”.*

To evaluate these data, which consistently show moderate effects, a Mann-Whitney test of entire dataset gathered in the time-course experiments (Supplementary Fig. 1a-c) shows that despite the modest effects *lat-1* loss of function has on overall and PH3-positive cells as well as on the M index at each given timepoint, the difference over the entire time-course are significant. Specifically, the p values are: progenitor zone 0.015; PH3-positive germ cells: 0.008; mitotic index: 0.032.

We hope these additions make the data more comprehensive and the interpretation clearer.

2. **Figure 2. The claim that *lat-1* is affecting GLP-1 signaling is not well supported. While a double mutant of two null alleles with no additive effect over each single mutant would suggest a possible linear pathway, that is not the case presented here and so the interpretation of the data presented in Figure 2 is problematic. GLP-1 Notch is thought to function in cell fate (stem versus differentiated), but not in proliferation rate. Typically, genes that act in the GLP-1 pathway cause premature differentiation (temporal or spatial) such that their double mutant combinations with a partial loss-of-function *glp-1* allele results in complete loss of the entire progenitor zone (e.g., *cye-1*). Conversely, genes that act in proliferation (e.g., *daf-2*, *eat-2*) cause changes in the mitotic profile (e.g., M-phase index) but not the same “*glp-1*-like phenotype” (loss of all progenitor cells). In these cases, the interpretation is that strong enhancement of the “*glp-1*-like phenotype” in the sensitized low-Notch signaling background is evidence for involvement in the pathway whereas a minor reduction in progenitor number is not.**

We agree with the reviewer that the loss of *lat-1* does not fully mimic the *glp-1* phenotype and that the effects observed in *lat-1* loss-of-function mutants display are considerably more subtle. We are aware of the challenges this presents and have therefore complemented the physiological data with functional analyses to strengthen our conclusions. Especially the Notch activation assay provides strong evidence for a direct link between LAT-1 and the Notch pathway (Fig. 2f, g), since it provides the most direct read-out of Notch activity as compared to e.g. PH3 cells or progenitor zone size. Together, the data give us the confidence to hypothesize that LAT-1 interacts with the Notch pathway. In this context, LAT-1 appears to play a modulatory role, fine-tuning rather than driving the pathway's activity.

However, we cannot exclude the possibility that LAT-1 has additional roles in the germ line. This notion is supported by the presence of LAT-1 in both germ cells and the distal tip cell (DTC). Therefore, the observed effects on germ cell proliferation might be influenced by other functions of LAT-1, such as potential roles in cell cycle regulation. In the revised manuscript, we have rephrased the relevant paragraph and removed terms such as “*glp-1*-like phenotype” to avoid overstatement.

To further explore the physiological interplay between *lat-1* and *lag-2*, we have also followed the reviewer's suggestion in comment 3 below and examined additional contexts in which *lat-1* may interact with *lag-2* (see our response below).

3. **Further support for the interaction with *lag-2* could be demonstrated with other *lag-2* phenotypes, such as the many interactions between *lag-2* and *lin-12*.**

We appreciate the reviewer's helpful comment and have therefore investigated the involvement of LAT-1 in additional *lag-2*-related phenotypes. Specifically, we selected two contexts in which Notch signaling via LAG-2 has been shown to play a role:

- **Rectum/anus formation during development:** During development, Notch signaling is (among others) essential for the formation of the rectum/anus. *lag-2* mutant nematodes do not form a proper rectum/anus, a process involving *lag-2*, *glp-1*, and *lin-12*¹. As *lat-1* has been shown to be expressed widely during development², we tested whether it might interact with the Notch signaling here. To this end, we assessed the presence or absence of the rectum and checked for the appearance of a protrusion at the site of the anal opening. We used *lat-1(ok1465)* and *lat-1(ok1465); lag-2(q420)* double mutants. Wild-type and *lag-2(q420)* individuals served as controls. We observed that *lat-1* mutants displayed a phenotype similar to *lag-2* mutants, though the absence of the anus occurred less frequently. The *lat-1; lag-2* double mutant was phenotypically indistinguishable from the *lag-2* single mutant. These findings have been added to the revised manuscript (Fig. 2h, i).

- Chemosensory avoidance of octanol: In the nervous system, Notch signaling controls chemosensory avoidance of octanol ³. As LAT-1 is also present on neurons ⁴, an interaction with LAG-2 in this context seems plausible. Therefore, we assessed the avoidance behavior of *lat-1(ok1465)* and *lat-1(ok1465); lag-2(q420)* double mutants compared to wild-type and *lag-2(q420)* individuals. The strain *osm-11(rt142)* was used as a positive control ³. Our analysis showed that both *lat-1* single mutants and *lat-1; lag-2* double mutants exhibited avoidance behavior similar to *lag-2* mutants. These results are presented in Fig. 2j of the revised manuscript.

In both contexts, *lat-1* mutants displayed phenotypes resembling those of *lag-2* mutants, and the double mutants did not show additive effects. These findings support the hypothesis that LAT-1 functions not only in the DTC to regulate germ cell proliferation via the Notch pathway, but also in other contexts. We hope these additional analyses further strengthen our argument for a functional interaction between LAT-1 and LAG-2.

4. Does LAT-1 also bind DTC-expressed APX-1 (at least in silico?), or generally to proteins with EGF-like extracellular motifs?

We thank the reviewer for this highly intriguing idea. To explore a potential interaction between LAT-1 and APX-1, we employed the same BRET-based cell culture assay previously used to analyze the LAT-1/LAG-2 interaction. Specifically, we fused APX-1 cDNA to Nanoluciferase (Nluc) to serve as the BRET donor, while Venus-tagged LAT-1 acted as the BRET acceptor. Our results show that LAT-1 does indeed interact with APX-1, as indicated by the BRET signal. These findings have been incorporated into the revised manuscript (Supplementary Fig. 7d), where we discuss the possibility that LAT-1 may participate in a broader interaction network with Notch ligands.

Notably, the five amino acids within the LAT-1 RBL domain and the seven within the GAIN domain, which have been shown to be critical for LAG-2 binding, also appear to contribute to the interaction with APX-1. This suggests that the underlying binding mechanism may be more general than initially assumed.

5. I) Figure 4. Demonstration of LAG-2 and LAT-1 co-localization on the DTC membrane would provide additional support.

We are grateful to the reviewer for this suggestion and have performed co-localization experiments. To investigate the spatial relationship between LAT-1 and LAG-2, we generated a *C. elegans* strain expressing *lat-1* tagged with a 5x V5 epitope in the second intracellular loop (a well-tolerated position that has shown not to alter receptor expression ²) as well as *lag-2::mTurquoise2* (allele *lag-2(bmd204 [lag-2::mTurquoise2^lox511]^2xHA)*) ⁵. Gonads from hermaphrodites staged at L4 + 8 hours were dissected, stained with antibodies, and analyzed using confocal and stimulated emission depletion (STED) microscopy (Reviewer Fig. R1).

Our imaging data reveal that both LAG-2 and LAT-1 are present on the distal tip cell (DTC), with LAT-1 also detected on germ cells. On the DTC, cellular colocalization of LAG-2 and LAT-1 is visible. However, we did not observe perfect one-to-one colocalization. This is likely due to differences in protein abundance - LAT-1 appears to be more abundant than LAG-2 - and potential variations in antibody affinity, which may affect the apparent stoichiometry. Complete colocalization is also not expected, as LAT-1 may have additional functions independent of LAG-2, particularly considering its presence on both germ cells and the DTC. Thus, while our findings demonstrate colocalization at the cellular level and partial overlap at subcellular resolution, we acknowledge that coincidental colocalization cannot be entirely ruled out. Therefore, although our data support spatial proximity between LAT-1 and LAG-2, definitive conclusions must be drawn with caution.

Reviewer Figure R1. LAT-1 and LAG-2 colocalize on the DTC. (a) Schematic depiction of LAT-1 and LAG-2 on the DTC and germ cells. LAT-1 is tagged with 5x V5 in the second intracellular loop and LAG-2 carries an mTurquoise2 at the C terminus. Both were detected using antibodies specific to the respective tag. (b) Confocal (left) and STED (center) analyses of LAT-1 and LAG-2 localization indicate presence at the membrane and partial colocalization of the two. Gonads were stained (primary antibodies: mouse anti V5, rabbit anti GFP; secondary antibodies: goat anti-rabbit Abberior star red, goat anti-mouse Alexa 568), and subjected to confocal microscopy (left) as well as stimulated emission depletion (STED) microscopy (center). Insets (right) show magnified clusters of colocalized LAT-1 and LAG-2. Excitation was at 561 nm and 640 nm with depletion at 775 nm. Images were taken with an Abberior Facility Line microscope, deconvolution of STED images was performed with Huygens Professional.

II) It is unclear why the assay was confined to the head.

The analyses using the BiFC assay (Fig. 4) were confined to the head region of the worms for technical reasons. In this assay, LAT-1 and LAG-2 were each fused to one half of a split Venus fluorescent protein, with expression driven by a heat-shock promoter. As this promoter is particularly active in the pharynx⁶, resulting in strong and easily detectable expression, we chose this region to monitor and quantify bimolecular fluorescence complementation (Fig. 4). The use of a heat-shock promoter was necessary because the endogenous expression levels of LAT-1 and LAG-2 were too low to yield robust BiFC signals. Moreover, overexpression of LAT-1 in the DTC and germ cells or of LAG-2 in the DTC could potentially cause detrimental effects. This is due to the nearly irreversible nature of Venus reconstitution, which stabilizes interactions that may otherwise be transient. We apologize that in the initial manuscript, it did not become clear why the assay was confined to the head. We apologize that in the initial manuscript, it did not become clear why the assay was confined to the head. We have now included an explanation in the legend of Fig. 4 (stating: “For signal monitoring, the pharynx was chosen as the heat-shock promoter is highly active in pharyngeal neurons.”) and in the main text (reading: “To determine whether the observed LAT-1/LAG-2 interaction occurs in vivo in *C. elegans*, Bimolecular Fluorescence Complementation (BiFC) was conducted. For that purpose, the two parts of a split Venus protein (VN/VC) were fused to one of the interaction partners each under the control of the heat-shock promoter *hsp16.41p*. Upon heat-shock, the proteins are produced, and a fluorescent signal occurs if they are in close enough proximity to interact. As the heat-shock promoter is very strong in pharyngeal neurons, both

proteins express at relatively high levels, allowing for a robust detection of a possible interaction even if it was weak. Thus, for monitoring potential signals, we assessed the pharynx.”).

The explanation for the use of the heat-shock promoter is also included in the Results section:

“It is important to note that, while for all other *in vivo* experiments involving modified *lat-1* the genomic locus in *C. elegans* was edited, in this particular case we utilized extrachromosomal arrays and an overexpression system based on heat-shock for two main reasons. First, endogenous LAT-1/LAG-2 interaction levels would likely be too low to detect. Secondly, the reconstitution of the split Venus fragment is nearly irreversible, thus stabilizing the LAT-1/LAG-2 interaction once it forms, might have detrimental effects.”

6. Figure 5. The modest effect on the germ line and the high level of variability in the data make it difficult to interpret the locus of action. In addition, the mitotic index, rather than number of PH3 positive cells, should be reported given that the strains vary in number of progenitor cells.

The reviewer is correct in noting that the effect on the germ line is not pronounced. To address this, we performed additional experiments assessing both overall germ cell numbers and the number of proliferating cells (e.g., Fig. 5c–e; initial manuscript: $n \geq 25$ in at least 6 independent experiments; revised manuscript: $n \geq 31$ in at least 7 independent experiments). While these additional data did not substantially increase the observed differences, they did help to strengthen the dataset and improve statistical significance in some cases. To determine the locus of LAT-1 action, we would like to highlight that the combination of the *in vivo* data showing that *lat-1* expressed exclusively in the DTC rescues the *lat-1* mutant phenotypes (Fig. 5) and the total absence of any signal in *in vitro* BRET experiments in *trans*, indicates the likeliness of this interaction occurring at the DTC membrane (i.e. in *cis*).

To determine the site of LAT-1 action, we combined different approaches to overcome the high variability in the data. As such, the *in vivo* data showing that *lat-1* expression restricted to the distal tip cell (DTC) is sufficient to rescue the *lat-1* mutant phenotypes (Fig. 5), combined with the complete absence of signal in *in vitro* BRET assays performed *in trans*, strongly suggest that the interaction likely occurs at the DTC membrane. These independent lines of evidence give us sufficient confidence in our conclusion.

Moreover, we have now calculated and included the mitotic indices for all experiments analyzing PH3-positive and total germ cell numbers (see Figs. 1h, 5e, Supplementary Fig. 1c, f).

Reviewer #2

- 1. A more complete scheme on the Notch signaling pathway (like the one shown in <https://www.sciencedirect.com/science/article/pii/S0092867409003821?via=ihub#fig1>) for the general readership of Nat. Com. would be useful to better understand the biological context of this work. The one in Figure 2a is insufficient, as it does not show important elements discussed in this study such as NCID, Notch receptor (not explicitly mentioned) or the LAT-1 and its discussed effect.**

We thank the reviewer for pointing out this lack of comprehensiveness in our original figure. We have taken the suggestion on board and have revised Fig. 2a to provide a more comprehensive overview of the Notch signaling pathway. Specifically, we have added the general molecule names besides the *C. elegans* ones (e.g. Notch receptor (GLP-1)). Additionally, we have incorporated further pathway details, such as the sequential cleavage events of the Notch receptor by specific enzymes and the downstream target genes. We included a brief description of the pathway not only in the respective figure legend, but also in the main manuscript text:

“In the C. elegans stem cell niche, the initial signal in this highly conserved cascade originates from the DTC, which expresses the DSL ligand lag-2. The interaction of this membrane-bound molecule with the Notch receptor GLP-1 on adjacent germ cells triggers the activation of GLP-1, which is cleaved twice with one cleavage by a γ secretase yielding the release of the Notch intracellular domain (NICD). This NICD acts as a transcription factor for the direct transcriptional targets (lst-1/sygl-1), which in turn activate repressors repression of gld-1-3 and other mitosis-inhibiting or meiosis-promoting factors (Fig. 2a).”

While Fig. 2a is intended to give a foundational overview of Notch signaling, Fig. 5f places LAT-1 within this signaling context. Accordingly, we have also updated Fig. 5f to reflect this, adding further information for clarity. We hope these revised figures with additional textual explanation enhance the clarity and completeness of the manuscript.

2. The authors predicted interactions between the extracellular domains of LAT-1 with LAG-2 and GLP-1, using AlphaFold2 Multimer. I find this structural model very intriguing but it also leaves many open questions. Did the authors investigate the model quality/stability by carrying out MD simulations?

We appreciate the reviewer's interest in the structural modeling of the investigated complex and agree that further computational analyses are important to assess the robustness of predicted protein-protein interactions. The reviewer is right, while AlphaFold2 Multimer provides valuable initial hypotheses on possible interaction interfaces, MD simulations offer complementary insights into the conformational stability and conformational flexibility of such complexes over time. To address this, we followed the reviewer's suggestion and performed molecular dynamics (MD) simulations of the predicted LAT-1/LAG-2 complex. In agreement with our predictions and experimental data we found reasonable conformational stability for the predicted complex. Accordingly, the MD simulations support our hypothesis/model.

The MD simulation data is now included in the revised manuscript in Supplementary Fig. 5 with an explanation in the respective figure legend and we have added the following to the results section:

“To validate the AlphaFold2 Multimer predicted complex and to investigate its conformational stability, we carried out all-atom molecular dynamics (MD) simulations. The structural integrity of the full LAG-2-GAIN-RBL complex was retained in all three repeats of 1-microsecond MD simulations. Within the simulation time, besides some thermal fluctuations of the GAIN and RBL domains against LAG-2, we did not observe major structural re-arrangements, validating the predicted complex (Supplementary Fig. 5).”

3. The binding interface of the GAIN domain seems to be larger compared to the RBL domain. Would this suggest that the GAIN domain contributes more to the binding?

The reviewer is absolutely right, the *in silico* models suggest a broader contact surface between the GAIN domain of LAT-1 and LAG-2 compared to the RBL domain. To provide more detailed insights, we quantified the interaction interface using Rosetta's Interface Analyzer and performed per-residue energy decomposition. The total predicted interaction energy was 64.3 Rosetta Energy Units (REUs), with the GAIN domain alone contributing 48.5 REUs. This suggests that the majority of the interaction likely arises from the GAIN domain.

However, it is important to note that REUs are computational estimations and do not directly correlate with experimental binding affinities. A larger predicted interface does not necessarily imply a greater functional contribution or physiological effect. In line with this, our mutational analyses (Fig. 3c) show that only the combined mutation of both RBL and GAIN domain residues significantly reduces binding affinity, while mutations in either domain alone had minimal or no

effect. This suggests that both domains may be required to establish a stable and specific interaction, potentially acting cooperatively.

It is possible that the GAIN domain engages in more unspecific interactions, which are harder to disrupt via single amino acid substitutions, whereas the RBL domain may provide more specificity through side-chain contacts. This could explain the discrepancy between the computationally predicted interface size and the experimental mutational sensitivity. We have added a respective paragraph in the discussion:

“Further support for this interaction is provided by BRET and BiFC analyses in both cells and live nematodes (Figs. 3, 4). It has to be noted that a larger interface does not necessarily equate to greater functional contribution and/or physiological effect. As such, our mutational analyses (Fig. 3c) indicate that only the combined mutation of both RBL and GAIN domain residues significantly reduces binding affinity, whereas single-domain mutations had minimal or no effect. This implies that both domains may be required to form a stable and specific interaction, possibly acting cooperatively.”

4. Overall, the reader would appreciate more information about binding interfaces, including interface size, amount of polar and hydrophobic interactions, etc.

We have now included the following additional information about the binding interface:

- Interface size: The buried surface area (BSA) at the LAT-1/LAG-2 interface is approximately 2768 Å² (calculated using Rosetta InterfaceAnalyzer).
- Polar vs. hydrophobic interactions: The interface comprises approximately 42% polar contacts and 58% hydrophobic contacts, while the GAIN domain has a slightly lower hydrophobic contact proportion than the RBL domain (57% vs. 61%). Notably, several key stabilizing residues in the RBL and GAIN domains participate in hydrogen bonding.
- Hydrogen bonds: The model predicts approximately 9 hydrogen bonds at the interface, most of which are concentrated in the region involving the LAT-1 GAIN domain with six.

To further strengthen our line of argumentation regarding the binding interface of LAT-1, we looked at the conservation of the residues forming the binding interface throughout the *Caenorhabditis* species (Supplementary Fig. 6). The respective sequences within both the RBL and the GAIN domain are highly conserved, indicating that also the binding could be conserved, too.

We believe that this information provides the reader with a clearer understanding of the interaction's physical and chemical nature. We thank the reviewer for encouraging us to strengthen this aspect of the manuscript.

5. The authors analyze alanine mutations of the RBL and GAIN domain. For instance, they find that the RBL or GAIN mutant alone does not significantly reduce the affinity of LAT-1 for LAG-2 (Figure 3c). Would this imply that one domain alone is sufficient to mediate the observed binding including the *in vivo* function? Can the authors carry out more experiments to investigate this possibility?

The reviewer is right, it is a well justified hypothesis that the mutations in one of the two domains might impair LAT-1 function although no reduction in the binding of LAT-1 to LAG-2 was observed. We investigated how mutations in the GAIN domain compare to those in the RBL domain in terms of their impact on receptor function. To this end, we introduced mutations based on the residues identified to be essential for the binding using CRISPR-Cas9 genome editing. Mutations in the RBL domain, either alone or in combination with GAIN domain mutations, produced only a very limited number of homozygous worms. This low yield was insufficient to establish a stable strain for further detailed analysis. However, this is not surprising, as the RBL domain has previously been shown by us and others to be critical for various LAT-1 functions. We have demonstrated that the receptor

lacking the RBL domain is generally non-functional in any context where LAT-1 is normally active⁷. Additionally, a recent study identified the Toll-like receptor homolog TOL-1 as a binding partner of LAT-1⁸. The interaction between LAT-1 and TOL-1 is mediated, at least in part, by the same region within the RBL domain that we identified as essential for binding to LAG-2. It is therefore plausible that the introduced mutations in the RBL domain disrupt not only the LAT-1/LAG-2 interaction but also other critical interactions, thereby compromising LAT-1 function more broadly.

In contrast, mutations in the GAIN domain (Reviewer Fig. R2a) resulted in homozygous worms. These did not exhibit any visible differences compared to wild-type controls, although a slight trend was observed. This was true both for the number of PH3-positive germ cells (Reviewer Fig. R2b) and for the overall number of germ cells in the proliferative zone (Reviewer Fig. R2c). These findings suggest that mutations in the GAIN domain alone do not significantly impair receptor function in this specific context. Nonetheless, given the relatively small phenotypic differences between *lat-1* mutants and wild-type worms, it might well be that a potential partial loss of function by the mutations in the GAIN domain cannot be resolved.

Reviewer Figure R2. Mutating solely the GAIN domain is not sufficient to result in a *lat-1* null mutant phenotype. a) We introduced mutations identified in Figure 3 in the genetic locus encoding *lat-1*. b) Representative images of PH3-stained gonads of wild-type, *lat-1* mutant and *lat-1*(GAIN^{mut}) worms. Asterisks indicate DTC and the dashed lines the end of proliferative zone. c), d) The amount of PH3-positive germ cells (c) and cells in the proliferative zone (d) of *lat-1*(GAIN^{mut}) worms are not significantly different from those in wild-types. Quantification of images shown in b), $n \geq 13$ in three independent experiments. n.s. = not significant, * $p < 0.05$, ** $p < 0.01$.

- In Figure 3a, the LAG-2 structural representation should be explicitly labeled as such to enhance clarity, and each element in the structure should be described in the caption (e.g.: “GAIN domain (yellow cartoon)”).**

We thank the reviewer for this helpful suggestion. We have updated Figure 3a to explicitly label the LAG-2 structure within the model, improving clarity and visual distinction between the two proteins. Further, we have highlighted the residues identified to potentially be essential for the interaction, these are highlighted as cartoon representation. In addition, we have revised the figure caption to describe each structural element, including domain annotations and color coding. The caption now reads:

“a) AlphaFold2 Multimer models suggest an interaction of the extracellular region of LAG-2 (blue) with both the RBL (yellow) and GAIN (cyan) domain of LAT-1. The interacting regions in LAT-1 are the main protruding loop of the RBL domain as well as a stretch of the GAIN domain. The HRM domain (grey) is not involved in binding. The interacting residues are highlighted as cartoon representation. Details on energy breakdown values as calculated by Rosetta per_residue_energies application and are given in Supplementary Fig. 4.”

We believe these changes significantly improve the readability and interpretability of the figure and thank the reviewer for pointing this out. As mentioned in our response to the reviewer's comment 7., we updated the model and figure accordingly.

7. Extended Data Figure 4 should add a label for the vertical left residues in both plots and explain what is in the top and bottom. In addition, the cell and x-axis numbers are quite small. Color scale range [0 to 1] does not match the numeric ranges of the figures [-71.1 to 1.7]

We thank the reviewer for their careful evaluation of Extended Data Figure 4 and agree with the suggested improvements. We have revised the figure (which is now labeled Supplementary Fig. 4) accordingly and addressed the following aspects:

Overall figure layout:

We have combined the figure in one final plot and improved the readability.

Font size and axis scaling:

Axis labels and tick marks have been enlarged for improved readability, especially the x-axis residue numbers and the cell text. These changes ensure the figure remains accessible when printed or viewed at smaller scales. Additionally, we changed the cutoff level from +/- 0.1 REU to +/- 0.2 REU to focus on more relevant residue interactions

Residue labels:

The y-axis (left side) of the plot now includes domain identifiers, clearly indicating which residues are being assessed for energetic contribution. We have added explanatory labels to clarify that the x-axis shows the per-residue interactions contributed by LAT-1 residues, while the y-axis shows the contribution of LAG-2 residues.

Color scale correction:

While we fully agree that a color scale should generally reflect the numerical range of the underlying data, in this case we intentionally chose not to use the absolute minimum and maximum REU values for the color mapping. Our goal was to visually emphasize the residues with moderate, yet biologically meaningful, energetic contributions rather than highlighting only the extreme outliers. Using the full range (e.g., from +1 to -70 REU) would have compressed the color resolution for residues contributing in the range of -1 to -2 REU, which we believe are important for interpreting the specificity and cooperativity of the interaction. By narrowing the color scale, we were able to better resolve these mid-level contributors, which are often overlooked but may play a key role in stabilizing transient or interface-spanning contacts.

Please note that during the revision process, we identified that one residue, glutamate 121 in the DSL domain of LAG-2, was not modeled accurately in the original AlphaFold2 Multimer predictions. Although this residue and its immediate neighboring residues do not participate in the LAT-1/LAG-2 interaction interface in any of the models, we opted to rerun all computational modeling and energy calculations to ensure the robustness of our data. The refined models confirmed the previously identified interaction interface and the key regions between LAT-1 and LAG-2, without any changes to the binding site composition. However, the local structure of the LAG-2 DSL domain was improved, adopting a more defined beta-barrel fold. As a result of these adjustments, we updated all figures based on the structural models, including Fig. 3 and Supplementary Fig. 4 (previous Extended Data Figure 4). While the overall conclusions remain unchanged, the updated visualization now more accurately reflects the refined structural context and interaction energies.

We thank the reviewer again for prompting a closer inspection of these figures and for helping us improve the clarity and accuracy of the presentation.

8. Line 182 add Fig 3b reference.

We have added the Figure reference in the revised version of the manuscript.

9. Extended Data Figure 5 uses Rosetta Energy Units (REU). Please provide more information on this unit.

We thank the reviewer for pointing out the need for clarification regarding Rosetta Energy Units (REUs). REUs are the internal scoring units used by the Rosetta software suite to estimate the energetic favorability of molecular conformations and interactions. They are calculated as a weighted sum of various physical and statistical energy terms, including van der Waals forces, solvation effects, hydrogen bonding, and electrostatics, as well as knowledge-based preferences for bond angles and torsions. Importantly, REUs do not represent absolute binding energies (e.g., in kcal/mol, but can be taken as computational estimation of kcal/mol) and are best interpreted comparatively rather than quantitatively. In the context of Extended Data Figure 5 (now termed Supplementary Fig. 4), REUs were used for per-residue decomposition of the LAT-1/LAG-2 interaction interface. More negative REU values indicate stronger stabilizing contributions from individual residues or domains, while values closer to zero reflect weaker or neutral contributions. To improve clarity, we have now included this explanation of REUs in brief in the figure caption of supplementary Fig. 4. We hope this helps readers interpret the data more accurately.

10. Figure 3 should be better described in the caption, for example adding the respective explanation with the coloring for the structures (Yellow, RBL domain, Grey surface LAG-2...etc.)

We thank the reviewer for this helpful suggestion. We have revised the figure legend to describe each structural element, including domain annotations and color coding. We believe these changes significantly improve the readability and interpretability of the figure and thank the reviewer for pointing this out. As mentioned in our response to the reviewer's comment 7.), we updated the model and figure accordingly.

11. HBD loop. the HBD abbreviation in Fig 3, is not explained. Also, this loop seems to be named differentially in "Supplementary Protocol Capture Molecular Modeling" as HRM, HormR, and HBD (?) in Figure 3. Please check.

We thank the reviewer for catching this inconsistency and apologize for any confusion caused by the use of different names for a single domain. The loop region corresponds to the Hormone Receptor Motif (SMART ACC: SM000008; Pfam: PF02793). The use of different terms arose from the fact that the Hormone Receptor Motif (HRM or HormR) is sometimes also referred to as the Hormone Binding Domain (HBD). To avoid confusion and maintain consistency, we have standardized the terminology throughout the manuscript, figures, and supplementary protocol. All references now consistently use the term HRM. Additionally, we have updated the caption of Fig. 3 to include a clear explanation. We appreciate the reviewer's attention to detail, which has helped us improve the clarity of our presentation.

12. The Method section "Molecular modeling of LAT-1 and LAG-2" and the "Supplementary Protocol Capture Molecular Modeling" are not clearly explained and should be revised. For example, the

order of the process changes in both explanations. A flowchart or figure would enhance the understanding of this section. Please revise.

We agree with the reviewer that the molecular modeling workflow would benefit from a clearer and more consistent presentation. To account for this, we have revised both the Methods section and the Supplementary Protocol to ensure consistent terminology and to present the modeling steps in a unified, logical order.

To further enhance clarity, we have added a schematic depiction that presents a step-by-step flowchart of the entire molecular modeling pipeline (Supplementary Protocol Capture Molecular Modeling). We believe this visual summary will help readers to follow the process more easily and appreciate the methodological rigor of the modeling and interaction analyses.

- 13. The authors mention: “While no interaction was predicted between LAT-1 and GLP-1, an interaction between LAT-1 and LAG-2 appeared likely, with the interaction likely occurring via the LAT-1 Rhamnose-binding lectin (RBL) and GPCR autoproteolysis-inducing (GAIN) domains (Fig. 3a, Extended Data Fig. 4).” Here, the word “likely” should be revised. Use better scores of the provided AlphaFold model.**

We appreciate the reviewer’s suggestion and understand the comment to provide a clearer and more confident statement based on computational score. We initially used the word “likely” to make sure that readers note that there are limitations to the models. While AlphaFold2 Multimer provides useful structural hypotheses, its confidence metrics, such as the pLDDT, ipTM and pTM scores, can be affected by unstructured regions and multi-domain architectures, which are present in both LAT-1 and LAG-2. These scores do not exclusively reflect the confidence of the protein–protein interface, but rather the overall model quality, including domain packing and intrinsic disorder. To avoid overstating the certainty of the interaction based solely on these computational predictions, we used the term “likely” to reflect the model’s hypothesis-generating role. However, based on the consistent domain-level contacts observed across multiple predicted structures and various domain combinations we have now rephrased the respective paragraph to make it more comprehensive:

“Based on the consistent domain-level contacts observed across multiple predicted structures and various domain combinations, the computational model highlighted a binding hypothesis with interactions between LAT-1 and LAG-2, but no convergence of binding interfaces between LAT-1 and GLP-1 was observed. The interaction with LAG-2 is mediated via the LAT-1 rhamnose-binding lectin (RBL) and GPCR autoproteolysis-inducing (GAIN) domains (Fig. 3a, Supplementary Fig. 4).”

REFERENCES

- 1 Lambie, E. J. & Kimble, J. Two homologous regulatory genes, *lin-12* and *glp-1*, have overlapping functions. *Development* **112**, 231-240 (1991).
- 2 Langenhan, T. *et al.* Latrophilin signaling links anterior-posterior tissue polarity and oriented cell divisions in the *C. elegans* embryo. *Dev Cell* **17**, 494-504 (2009). <https://doi.org/10.1016/j.devcel.2009.08.008>
- 3 Singh, K. *et al.* *C. elegans* Notch signaling regulates adult chemosensory response and larval molting quiescence. *Curr Biol* **21**, 825-834 (2011). <https://doi.org/10.1016/j.cub.2011.04.010>
- 4 Matus, D., Post, W. B., Horn, S., Schoneberg, T. & Promel, S. Latrophilin-1 drives neuron morphogenesis and shapes chemo- and mechanosensation-dependent behavior in *C. elegans* via

- a trans function. *Biochem Biophys Res Commun* **589**, 152-158 (2022).
<https://doi.org:10.1016/j.bbrc.2021.12.006>
- 5 Medwig-Kinney, T. N., Sirota, S. S., Gibney, T. V., Pani, A. M. & Matus, D. Q. An in vivo toolkit to visualize endogenous LAG-2/Delta and LIN-12/Notch signaling in *C. elegans*. *MicroPubl Biol* **2022** (2022). <https://doi.org:10.17912/micropub.biology.000602>
 - 6 Stringham, E. G., Dixon, D. K., Jones, D. & Candido, E. P. Temporal and spatial expression patterns of the small heat shock (hsp16) genes in transgenic *Caenorhabditis elegans*. *Mol Biol Cell* **3**, 221-233 (1992). <https://doi.org:10.1091/mbc.3.2.221>
 - 7 Prömel, S. *et al.* The GPS motif is a molecular switch for bimodal activities of adhesion class G protein-coupled receptors. *Cell Rep* **2**, 321-331 (2012). <https://doi.org:10.1016/j.celrep.2012.06.015>
 - 8 Carmona-Rosas, G. *et al.* Structural basis and functional roles for Toll-like receptor binding to Latrophilin adhesion-GPCR in embryo development. *bioRxiv* (2023). <https://doi.org:https://doi.org/10.1101/2023.05.04.539414>

Notch activity is modulated by the aGPCR Latrophilin binding the DSL ligand in *C. elegans*

Response to Reviewers' comments

We thank the reviewers for the overall positive feedback and the helpful comments on our manuscript. In response to their suggestions, we have performed additional experiments to further support and strengthen our findings. Moreover, we have incorporated their constructive input to improve the clarity and overall impact of the study.

Below, we address all reviewer comments in detail and quote statements from the reviewers in **bold face**.

Reviewer #1

In general, the authors have addressed this reviewer's major concerns. However, there are still several potentially misleading that need to be addressed:

- 1. Line 109: "a decreased number of proliferating cells and an altered proliferative zone size have been observed in *glp-1* loss-of-function mutants". First, there are no proliferating germ cells in "*glp-1* loss-of-function" mutants. The authors are referring to reduction-of-function scenarios in temperature sensitive (ts) mutants. Second, the "proliferative zone" was renamed within the field over the last several years as "progenitor zone" to conform with the wider stem cell biology field. This term should be replaced throughout the manuscript and figures.**

We are very grateful to the reviewer for pointing out these important aspects. We have altered the description of the *glp-1* strain from "loss-of-function" to "reduction-of-function" throughout the entire manuscript. Similarly, we have exchanged the term "proliferative zone" with "progenitor zone" in the text and the figures.

- 2. Line 129: "Similar to *lat-1* mutants, hermaphrodites carrying single loss-of-function mutations in *lag-2* (*lag-2(q420)* 31) and *glp-1* (*glp-1(bn18)* 32) exhibit defects in germ cell proliferation and meiotic entry..."**

And Line 329: phenotype similar to that of *glp-1* loss-of-function mutants (allele *bn48*)"

Neither *lag-2(q420)* nor *glp-1* (*glp-1(bn18)*) are "loss-of-function" alleles. They are both ts reduction-of-function alleles. This should be clarified. And, in 329, the allele is presumably *bn18*?

We thank the reviewer for highlighting this inaccuracy and have now described the respective *glp-1* and *lag-2* alleles as "reduction-of-function" alleles (see also our response to reviewer comment 1).

The reviewer is correct, the allele referred to in line 329 of the manuscript is *bn18*. We apologize for this typo and have corrected it.

- 3. Line 391: "However, only subsequent ligand endocytosis truly activates the receptor by generating a mechanical force that promotes a conformational change within the Notch receptor."**

Mechanical force on the DSL ligand is not thought to be a required mechanism for Notch receptor activation in *C. elegans* (see Langridge et al. 2022)

We apologize for the misleading text passage and have adjusted the respective paragraph in our manuscript to highlight the fact that the force exerted by the Notch ligand required to activate the Notch receptor is potentially not triggered by Epsin-mediated endocytosis in *C. elegans* and hence, can be much less strong than in other organisms such as *Drosophila melanogaster*. Instead, pure membrane tethering seems to be sufficient. The passage now reads:

*“Alternatively, LAT-1 could increase the effect of LAG-2 on GLP-1. It is well established that for the activation of the Notch pathway, the Notch ligand binds to the Notch receptor on opposing cells (summarized in ^{1,2}). Subsequently, a certain level of force exerted from the Notch ligand is required to activate the receptor, which leads to the exposure of protease recognition sites and receptor cleavage at two sites, one extracellularly and one in the transmembrane domain that releases the NICD (summarized in ^{1,2}). In *Drosophila melanogaster* and vertebrates, this mechanical force needs to be quite strong and is generated by ligand endocytosis, promoting a conformational change within the Notch receptor. In *C. elegans*, much lower force thresholds are in place, tethering the ligand to the membrane is sufficient ³. A hypothesis could be that LAT-1 acts as an allosteric modulator, enhancing the conformational changes in the LAG-2/GLP-1 complex required for signaling.”*

Reviewer #2

- 1. The authors have addressed all of my comments satisfactorily and have substantially improved the manuscript in the revised version. I have no further concerns.**

We thank the reviewer for taking the time to review our manuscript.

REFERENCES

- 1 Kopan, R. & Ilagan, M. X. The canonical Notch signaling pathway: unfolding the activation mechanism. *Cell* **137**, 216-233 (2009). <https://doi.org:10.1016/j.cell.2009.03.045>
- 2 Suarez Rodriguez, F., Sanlidag, S. & Sahlgren, C. Mechanical regulation of the Notch signaling pathway. *Curr Opin Cell Biol* **85**, 102244 (2023). <https://doi.org:10.1016/j.ceb.2023.102244>
- 3 Langridge, P. D., Garcia Diaz, A., Chan, J. Y., Greenwald, I. & Struhl, G. Evolutionary plasticity in the requirement for force exerted by ligand endocytosis to activate *C. elegans* Notch proteins. *Curr Biol* **32**, 2263-2271 e2266 (2022). <https://doi.org:10.1016/j.cub.2022.03.025>